# Aurora kinase A promotes trained immunity via regulation of endogenous S-adenosylmethionine metabolism

**Mengyun Li[1,2], Huan Jin[2], Yongxiang Liu[3], Zining Wang[2], Lin Li[2], Tiantian Wang[2], Xiaojuan Wang[2], Hongxia Zhang[4,5,6], Bitao Huo[7], Tiantian Yu[7], Shoujie Wang[7], Wei Zhao[8], Jinyun Liu[2,7,9], Peng Huang[2,7], Jun Cui[1]\*, Xiaojun Xia[2,9]\***

[1]MOE Key Laboratory of Gene Function and Regulation, Guangdong Province Key Laboratory of Pharmaceutical Functional Genes, State Key Laboratory of Biocontrol, School of Life Sciences, Sun Yat-sen University, Guangzhou, China; [2]State Key Laboratory of Oncology in South China, Guangdong Provincial Clinical Research Center for Cancer, Sun Yat-sen University Cancer Center, Guangzhou, China; [3]Guangzhou National Laboratory, Guangzhou International Bio-Island, Guangzhou, China; [4]Department of Pathology, School of Basic Medical Sciences, Southern Medical University, Guangzhou, China; [5]Department of Pathology, Nanfang Hospital, Southern Medical University, Guangzhou, China; [6]Guangdong Provincial Key Laboratory of Molecular Tumor Pathology, Guangzhou, China; [7]Metabolic Center, Zhongshan School of Medicine, Sun Yat-sen University, Guangzhou, China; [8]Center for Stem Cell Biology and Tissue Engineering, Key Laboratory for Stem Cells and Tissue Engineering, Ministry of Education, Sun Yat-sen University, Guangzhou, China; [9]Hainan Academy of Medical Sciences, Hainan Medical University, Haikou, China

**\*For correspondence:**
cuij5@mail.sysu.edu.cn (JC);
xiaxj@sysucc.org.cn (XX)

**Competing interest:** The authors declare that no competing interests exist.

## eLife Assessment

The authors use a range of techniques to examine the role of Aurora Kinase A (AurA) in trained immunity. The study is hypothesis driven, it uses **solid** experimental approaches, and the data are presented in a logical manner. The findings are **valuable** to the trained immunity field because they provide an in-depth look at a common inducer of trained immunity, beta-glucan.

**Abstract** Innate immune cells can acquire a memory phenotype, termed trained immunity, but the mechanism underlying the regulation of trained immunity remains largely elusive. Here, we demonstrate that inhibition of Aurora kinase A (AurA) dampens trained immunity induced by β-glucan. ATAC-seq and RNA-seq analysis reveal that AurA inhibition restricts chromatin accessibility of genes associated with inflammatory pathways such as JAK-STAT, TNF, and NF-κB pathways. Specifically, AurA inhibition promotes nuclear localization of FOXO3 and the expression of glycine N-methyltransferase (GNMT), a key enzyme responsible for S-adenosylmethionine (SAM) consumption. Metabolomic analysis confirms a reduction in SAM level upon AurA inhibition. As a result of SAM deficiency, trained mouse macrophages exhibit decreased H3K4me3 and H3K36me3 enrichment on gene regions of *Il6 and Tnf*. Additionally, the tumor inhibition effect of β-glucan is notably abolished by AurA inhibition. Together, our findings identify an essential role of AurA in regulating trained immunity via a methylation-dependent manner by maintaining endogenous SAM levels through the mTOR-FOXO3-GNMT axis.

## Introduction

Trained immunity is depicted as a memory state of an innate immune cell, independent of adaptive immunity (*Naik and Fuchs, 2022*; *Ochando et al., 2023*). It can be induced by *Candida albicans*, Bacillus Calmette-Guérin vaccine, or microbial components such as β-glucan (*Cirovic et al., 2020*; *Kalafati et al., 2020*; *Quintin et al., 2012*). Trained immune cells exhibit a rapid and enhanced response to secondary related or unrelated stimulus, relying on chromatin remodeling and metabolic rewiring (*Bekkering et al., 2018*; *Cirovic et al., 2020*; *Su et al., 2021*). In trained cells, histone modification associated with active transcription, such as H3-histone-lysine-4 trimethylation (H3K4me3) is enriched in promoter regions of genes encoding inflammatory cytokines, supporting fast gene induction in response to secondary stimulation (*Su et al., 2022*). Trained cells typically undergo metabolic rewiring characterized by enhanced glycolysis, while the oxidative phosphorylation level shows context-dependent variations (*Cheng et al., 2014*; *Keating et al., 2020*). Intermediate metabolites in tricarboxylic acid (TCA) cycle, such as fumarate and succinate function as DNA or histone demethylase inhibitors (*Arts et al., 2016*), whereas methionine and SAM in one-carbon metabolism act as substrates for methyltransferases (*Ampomah et al., 2022*; *Yu et al., 2019*). Therefore, metabolic processing and metabolites intertwined with epigenetic regulation by directly supplying methylation substrates or regulating methylation enzyme activities. SAM, a ubiquitous methyl donor in biological process, can be converted into S-adenosylhomocysteine (SAH) for the synthesis of cysteine in glutathione metabolism. The ratio of SAM/SAH is critical in support of histone H3 trimethylation at lysine 36(H3K36me3) to enhance IL-1β transcription (*Yu et al., 2019*). However, it has not been clear how SAM level is regulated in the trained immunity process.

AurA, a serine/threonine kinase, plays a critical role in mitosis and is frequently overexpressed in tumor tissues. Inhibition of AurA in tumor cells restricts cell proliferation, migration, and induces cell death (*Donnella et al., 2018*; *Jingtai et al., 2023*; *Tham et al., 2024*; *Wang-Bishop et al., 2019*). Despite the well-established role of AurA on tumor cells' growth and tumorigenesis, its function in innate immune cells like macrophages or in inflammation is not well understood (*Ding et al., 2015*). A previous study reports that alisertib, a specific inhibitor of AurA, promotes gene expression of demethylase *KDM6B* during THP-1 cell (a human monocyte-like cell line) differentiation (*Park et al., 2018*). Whether AurA plays a similar role in regulating trained immunity in macrophages remains unknown. Trained immunity can lead to aberrant inflammatory activity as well as enhanced anti-tumor effect (*Wang et al., 2023b*). Understanding the role of AurA in trained immunity would enable us to further exploit AurA for clinical application in cancer therapy.

In this study, we screened an epigenetic drug library and found that AurA inhibitors notably attenuated the trained immunity induced by β-glucan. Genetic disruption of AurA also showed a similar inhibitory effect on trained immunity. Mechanistically, AurA inhibition reduced the activation of mTOR signaling, thus leading to the nuclear localization of transcription factor FOXO3. Nuclear FOXO3 promoted the expression of GNMT to decrease the intracellular level of SAM, thereby inhibiting H3K4me3 and H3K36me3 enrichment on the promoters of inflammatory genes such as *Il6* and *Tnf*. Moreover, pretreatment with alisertib abolished the anti-tumor effect conferred by β-glucan-induced trained immunity in vivo. Thus, we conclude that AurA is an essential kinase required for the epigenetic regulation in macrophages and antitumor activity of trained immunity.

## Results

### Inhibition of Aurora kinase A suppresses trained immunity in macrophages

Trained immunity is orchestrated by epigenetic reprogramming, while the specific epigenetic modulators involved in regulating trained immunity remain incompletely understood. To uncover the key epigenetic modulators in trained immunity, we performed a small molecule inhibitor screening by using an epigenetic drug library (*Figure 1—figure supplement 1A*, *Supplementary file 1*). Briefly, bone marrow-derived macrophages (BMDMs) were trained by β-glucan in the presence of different inhibitors for 24 hr, rested for 3 days, and then restimulated by LPS for secondary response. As trained immunity in macrophages strongly induces cytokines production including IL-6 and TNF upon restimulation by pathogen- or damage-associated molecular patterns (PAMPs or DAMPs) (*Chakraborty et al., 2023*; *Netea et al., 2020*), we used IL-6 level from the supernatant as a readout in our screening. The

targets of the epigenetic drug library were primarily categorized into several major classes, including Aurora kinase family, histone methyltransferase and demethylase (HMTs and KDMs), acetyltransferase and deacetylase (HDACs and SIRTs), JAK-STAT kinase family, AKT/mTOR/HIF, PARP family, and BRD family (*Figure 1—figure supplement 1B*). Consistent with previous studies, the drug screening results indicated that most inhibitors targeting PI3K/AKT/mTOR and HIF1α exhibited inhibitory effects on trained immunity (*Cheng et al., 2014*; *de Graaf et al., 2021*; *Sohrabi et al., 2018*). Furthermore, cyproheptadine, an inhibitor of SETD7, also suppressed trained immunity in our screening system (*Keating et al., 2020*). Among the total of 305 inhibitors in the compound library, 8 AurA inhibitors showed inhibitory effect on trained immunity-induced IL-6 production in multiple rounds of screening with different concentrations (*Figure 1—figure supplement 1C*). To further confirm whether AurA identified in our screening could modulate trained immunity, we optimized β-glucan training protocol and used three AurA-specific inhibitors, and found that all these AurA inhibitors significantly inhibited IL-6 production in β-glucan-trained mouse macrophages (*Figure 1—figure supplement 1D*). Among these inhibitors, alisertib is an orally active and highly selective inhibitor for AurA and has been applied in preclinical investigation (*Bavetsias and Linardopoulos, 2015*; *Mossé et al., 2019*; *O'Connor et al., 2019*). We further observed that alisertib obviously downregulated IL-6 and TNF in trained BMDMs in a concentration-dependent manner without affecting cell viability (*Figure 1A and B*). Moreover, alisertib also decreased the transcriptional level of *Il6* and *Tnf* in trained BMDMs (*Figure 1C*). Furthermore, the phosphorylation of AurA was increased by β-glucan but was blocked by alisertib (*Figure 1D*). To further confirm the role of AurA in trained immunity, we knocked down the expression of AurA by small interfering RNAs (siRNAs) in BMDMs (*Figure 1E*). Consistently, knocking down of AurA also inhibited the production levels of IL-6 and TNF in trained BMDMs (*Figure 1F*). Consistent with a recent report showing that tumor cells or tumor cell culture supernatant could function as the second stimulus for trained BMDMs (*Ding et al., 2023*), we also observed increased TNF and IL-6 production of trained macrophages upon secondary stimulation using MC38 tumor cells culture supernatant, and such effect was also inhibited by AurA knockdown or inhibition (*Figure 1G*). To verify whether AurA regulates trained immunity in different cell models, we trained J774A.1 cells as well as THP-1 cells with β-glucan and also observed a reduction of IL-6 and TNF levels under AurA knockdown or inhibition (*Figure 1—figure supplement 1E*). Moreover, administration of alisertib together with β-glucan also attenuated trained immunity in vivo (*Figure 1H and I*). Collectively, these results suggest that AurA inhibition suppresses β-glucan-induced trained immunity both in vitro and in vivo.

## Aurora kinase A inhibition remodels chromatin landscape of inflammatory genes

Epigenetic reprogramming towards an open chromatin status is considered the basis for innate immune memory (*Arts et al., 2016*; *Jeljeli et al., 2019*; *Moorlag et al., 2024*; *Novakovic et al., 2016*). To clarify the status of chromatin landscape under AurA inhibition, we performed an assay for transposase-accessible chromatin with high-throughput sequencing (ATAC-seq) in trained BMDMs. In detail, mice were trained by intraperitoneal injection with β-glucan with or without alisertib, and the bone marrow cells were isolated and differentiated into BMDMs for ATAC-seq analysis (*Figure 2—figure supplement 1A*). Principal component analysis showed divergent distribution between AurA-treated and -untreated trained BMDMs (*Figure 2A*). The erased regions by AurA inhibition were mapped into known AurA-regulated cellular processes like 'regulation of growth' (*Willems et al., 2018*), and inflammatory response-related processes like 'myeloid leukocyte activation' and 'MAPK cascade' (*Figure 2B*). Analysis of enriched transcriptional binding motifs indicated that the erased peaks by alisertib were transcriptionally regulated by IRF1/2, FOS, and STAT1/2, while the written peaks were transcriptionally regulated by PPARG, KLF4, ELF4, and FOXO1/3 (*Figure 2C*). Recent report shows that PPARG and KLF4 were critical transcription factors in controlling genes of macrophage M2 polarization and FOXO3 regulates the expression of multiple genes involved in pathways such as anti-inflammation and oxidative stress resistance (*Allemann et al., 2023*). Consistent with the reduced production of IL-6 and TNF proteins by alisertib, we observed a decreased chromatin accessibility of proinflammatory genes such as *Il6*, *Tnf*, *Cxcl2*, and *Il1a* by alisertib intervention (*Figure 2D*). In contrast, the peaks of genes encoded M2 marker *Mrc1* as well as *Chil3* were enhanced (*Figure 2E*). These results support that AurA regulates epigenetic

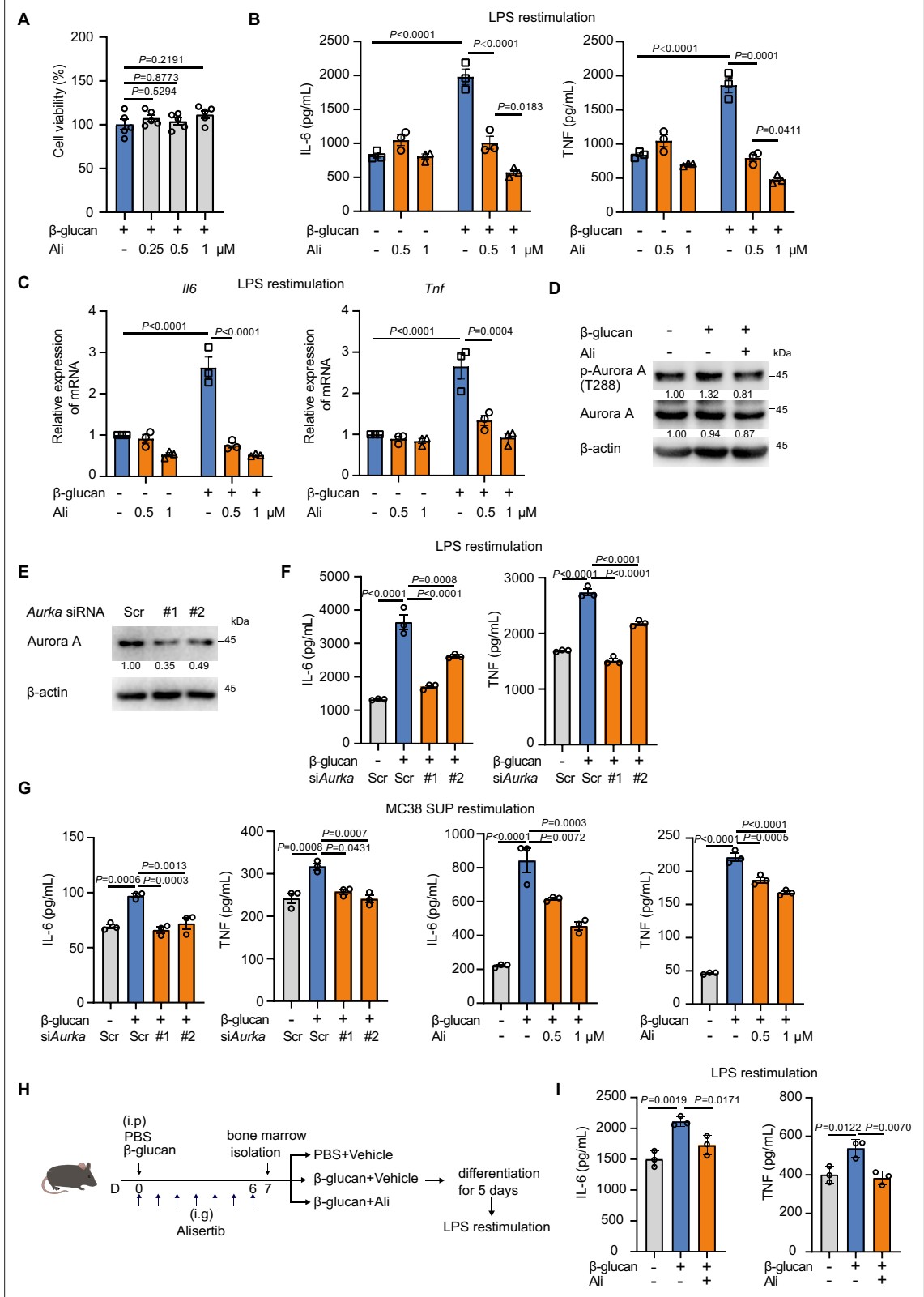

**Figure 1.** Inhibition of Aurora kinase A suppresses trained immunity in macrophages. (**A**) Bone marrow-derived macrophages (BMDMs) were trained with β-glucan at a dosage of 50 µg/mL in the presence of different concentrations of alisertib for 24 hr. The viability of BMDMs was measured by CCK8 assay. (**B**) Supernatant levels of IL-6 (left) and TNF (right) in trained BMDMs with alisertib (0.5 µM or 1 µM), followed by restimulation with LPS (100 ng/mL) for 24 hr. (**C**) qPCR analysis of relative mRNA expression of *Il6* and *Tnf* in trained BMDMs with alisertib (0.5 µM or 1 µM), followed by restimulation

*Figure 1 continued on next page*

*Figure 1 continued*

with LPS (100 ng/mL) for 6 hr. *Actb* served as a reference gene. (**D**) Immunoblotting analysis of Aurora kinase A (AurA) phosphorylation after the treatment of β-glucan (50 μg/mL) with or without alisertib (1 μM) for 90 min. (**E**) Immunoblotting analysis of AurA in BMDMs transfected with siRNA targeting AurA for 48 hr. (**F**) The BMDMs were firstly transfected with siRNAs for 48 hr and then stimulated with β-glucan (50 μg/mL). Supernatant levels of IL-6 and TNF were detected by ELISA after 3 days rest and restimulation with LPS (100 ng/mL) for 24 hr. (**G**) The BMDMs was stimulated with β-glucan (50 μg/mL) together with AurA knockdown or alisertib, followed by a rest for 3 days and restimulation with MC38 cell culture supernatant for 48 hr. (**H**) Graphical outline of in vivo training model (3 mice per group). Intraperitoneal injection, i.p; intragastric adminstration, i.g. (**I**) Supernatant levels of IL-6 (left) and TNF (right) in trained BMDMs as shown in (**H**). Data are presented as the mean ± SEM (except mean ± SD for in vivo training in I). *P* values were derived from one-way ANOVA with Dunnett's multiple-comparison test (**A, F, G, I**), compared with only β-glucan stimulation group; or two-way ANOVA with Tukey's multiple-comparison test (**B, C**), compared with every other group. In **D**, similar results were obtained from three independent experiments. Related to *Figure 1—figure supplement 1*, *Figure 1—source data 1–2*.

The online version of this article includes the following source data and figure supplement(s) for figure 1:

**Source data 1.** Uncropped and labeled blots for *Figure 1D and E*.

**Source data 2.** Raw unedited blots for *Figure 1D and E*.

**Figure supplement 1.** Targeting aurora A inhibits β-glucan-induced trained immunity.

---

changes in β-glucan-induced trained immunity and its inhibition restricts the chromatin accessibility for inflammatory genes in macrophages.

Meanwhile, we also performed RNA-seq with the trained BMDMs after LPS restimulation. Analysis of the transcriptome revealed that alisertib inhibited genes associated with pro-inflammatory pathways, including 'JAK-STAT signaling pathway,' 'TNF signaling pathway' as well as 'NF-kappa B pathway' in trained BMDMs (*Figure 2F*). In contrast, the up-regulated genes by alisertib were enriched in anti-inflammation pathways, such as 'FOXO signaling pathway' (*Figure 2G*). Additionally, differentially expressed transcription factors were analyzed and mapped into Gene Ontology (GO) with enrichment in 'negative regulation of Toll-like receptor signaling pathway,' 'negative regulation of NLRP3 inflammasome complex assembly,' and 'negative regulation of interleukin-6 production' (*Figure 2—figure supplement 1B and C*). Moreover, multiplex chemokine/cytokines array showed that alisertib decreased the chemokine/cytokines in trained BMDMs, such as IL-6, TNF, CXCL2, and IL-1α (*Figure 2—figure supplement 1D*). Collectively, these results demonstrate that alisertib restricts chromatin accessibility in genes associated with inflammation activation in trained BMDMs.

## Alisertib inhibits glycolysis and remodels TCA cycle

Various studies show that metabolic rewiring, such as glycolysis is critical for macrophage memory both in meeting energy demands and epigenetic modification (*Bekkering et al., 2018*; *Bhargavi and Subbian, 2024*; *Liu et al., 2024*). To determine the metabolic pathway that AurA may regulate in trained immunity, we analyzed the glucose metabolism in trained BMDMs. As expected, β-glucan increased glycolysis as indicated by higher extracellular acidification rate (ECAR), which was inhibited by alisertib, but no obvious change in oxidative phosphorylation level was detected (*Figure 3A–B*, *Figure 3—figure supplement 1A*). In accordance with enhanced glycolysis, U-$^{13}$C-glucose tracer showed that β-glucan treatment upregulated the glucose incorporation into lactate, and this induction was significantly inhibited by alisertib (*Figure 3—figure supplement 1B*). Meanwhile, metabolites in TCA cycle, including malate, citrate, α-KG, fumarate, and succinate, were all decreased in alisertib-treated trained BMDMs (*Figure 3—figure supplement 1B*). Previous study showed that fumarate was accumulated in β-glucan-treated cells, and it induced epigenetic reprogramming to facilitate trained immunity (*Arts et al., 2016*). However, alisertib treatment did not affect fumarate accumulation induced by β-glucan (*Figure 3C*), but increased tyrosine level (*Figure 3—figure supplement 1C*). As a recent report shows tyrosine could be interconverted to fumarate (*Li et al., 2023*), we speculated the decreased fumarate from TCA cycle by alisertib may be compensated from tyrosine metabolism. Taken together, AurA inhibition inhibits glycolysis and remodels glucose metabolism in trained immunity.

## Aurora kinase A regulates S-adenosylmethionine level in trained BMDMs

Next, we planned to identify the specific metabolites that support the function of AurA in trained immunity. KEGG analysis by using the differently expressed genes (DEGs) showed that glutathione

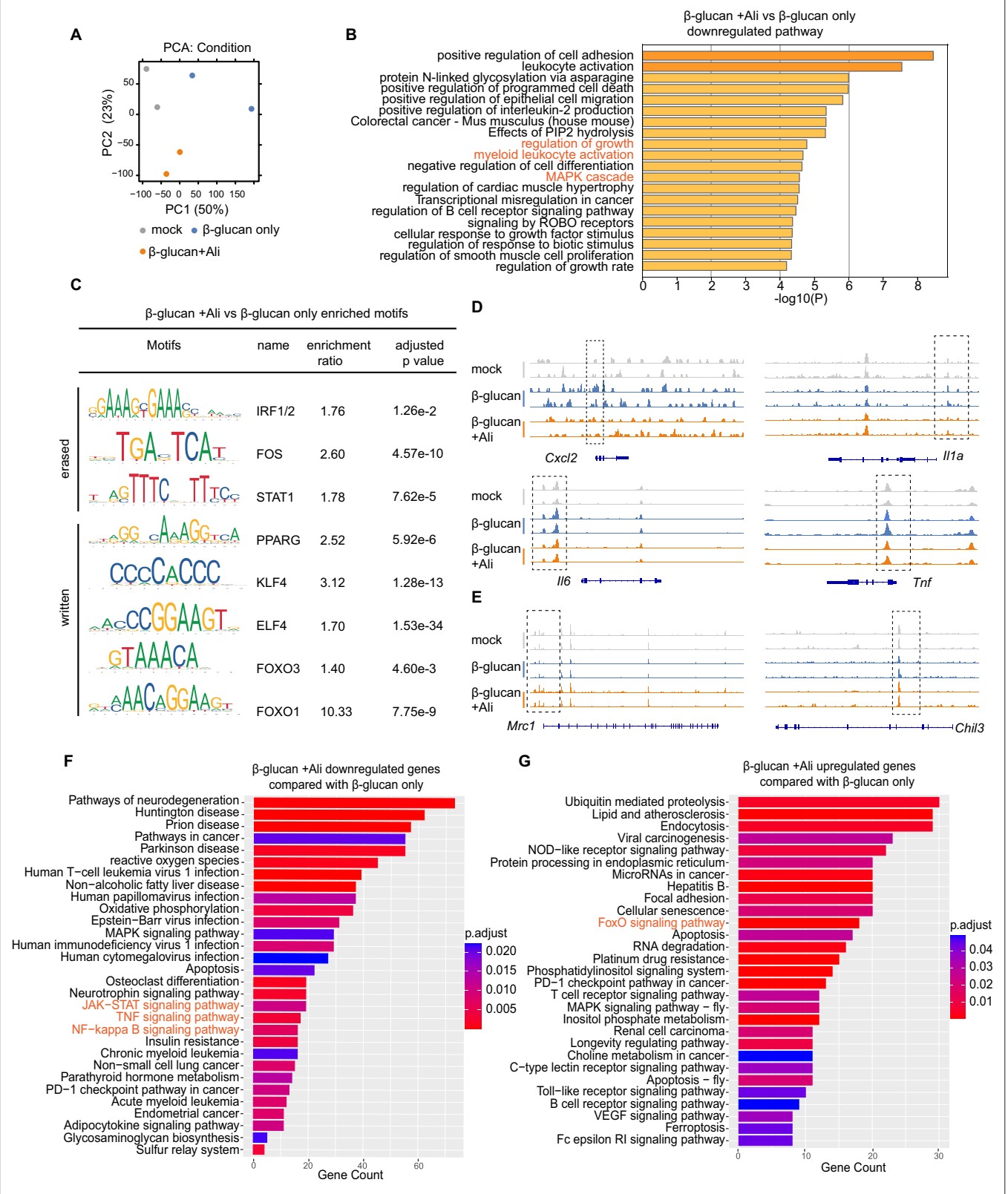

**Figure 2.** Aurora kinase A inhibition remodels chromatin landscape of inflammatory genes. (**A**) Principal component analysis (PCA) of gene peaks in ATAC-seq (n=2 mice per group). (**B**) Gene ontology (GO) enrichment analysis of erased peaks by alisertib in trained bone marrow-derived macrophages (BMDMs). (**C**) Representative motifs in the erased (n=15,431) and written (n=19,733) peaks, respectively. (**D, E**) Genome browser views of ATAC-seq signal of representative genes inhibited by alisertib, including *Cxcl2, Il1a, Tnf,* and *Il6* (**D**) and representative genes enhanced by alisertib, including *Mrc1*

*Figure 2 continued on next page*

*Figure 2 continued*

and *Chil3* (**E**). (**F, G**) KEGG enrichment of differentially expressed genes in trained BMDMs rechallenged with LPS; alisertib downregulated genes (**F**) and upregulated genes (**G**) were mapped into KEGG, respectively. Related to *Figure 2—figure supplement 1*.

The online version of this article includes the following figure supplement(s) for figure 2:

**Figure supplement 1.** Aurora kinase A inhibition suppresses the expression of transcription factors involved in inflammation activation.

(GSH) metabolism was significantly enriched (*Figure 3D*). We further found that the intracellular level of GSH in trained BMDMs was notably reduced after AurA inhibition (*Figure 3E*). It has been reported that β-glucan induces a modified cellular redox status as elevation of ROS level and enhanced synthesis of GSH (*Ferreira et al., 2021*; *Ferreira et al., 2023*). Genetic deletion of genes involved in GSH synthesis disrupts the cellular redox balance and dampens trained immunity (*Su et al., 2021*). To investigate whether AurA functions on trained immunity directly through decreasing GSH, we measured the level of serine and SAM, both of which are precursors of GSH (*Figure 3—figure supplement 1D*). By targeted liquid chromatography-tandem mass spectrometry (LC-MS) analysis, we found a significant reduction of SAM level under Aura inhibition, while the serine level remained unchanged (*Figure 3F*). Next, we further detected the downstream products of SAM, such as SAH and HCY. Consistently, we observed that alisertib also significantly decreased SAH and HCY in trained BMDMs (*Figure 3G*). The decreased intracellular level of these precursors of GSH indicated that GSH was probably not a metabolite that AurA directly acted on, and its reduction was a result from SAM deficiency. SAM is directly linked to epigenetics as a methyl donor. For example, SAM induction upregulates the total intracellular level of histone lysine 36 trimethylation (H3K36me3) and thus increases H3K36me3 modification at the gene region of *Il1b* to promote IL-1β expression (*Rodriguez et al., 2019*; *Yu et al., 2019*). Since AurA inhibition resulted in intracellular SAM reduction, we wondered whether SAM upregulation would rescue trained immunity inhibited by alisertib or by AurA knocking down. GNMT is a key enzyme responsible for the conversion from SAM to SAH, and its deficiency blocked the conversion from SAM to SAH and thus leading to the increased ratio of SAM to SAH (*Hwang et al., 2021*; *Li et al., 2015*; *Yen et al., 2013*). Furthermore, we found that AurA inhibition upregulated GNMT protein level in trained BMDMs (*Figure 3H*). To verify whether intracellular SAM level was responsible for AurA-mediated trained immunity, we knocked down GNMT in BMDMs (*Figure 3I*), and found that knockdown of GNMT increased the level of SAM while decreasing the level of SAH, leading to an increased ratio of SAM/SAH under AurA inhibition (*Figure 3J*). Furthermore, knockdown of GNMT rescued the IL-6 and TNF production in trained immunity under AurA inhibition in response to MC38 cell culture supernatant and LPS (*Figure 3K*). These results suggest that AurA promotes trained immunity by supplying endogenous SAM level which is controlled by GNMT.

## Inhibition of Aurora kinase A impairs histone trimethylation at H3K4 and H3K36

SAM is a well-known methyl donor for nearly all cellular methylation events, including DNA, RNA, and histone methylation (*Keen and Taylor, 2004*). Recent study reports that SAM deficiency limits histone methylation by phosphorylation of Rph1, which is a demethylase for H3K36me3 (*Ye et al., 2019*). Meanwhile, SAM prefers to induce histone methylation changes in targeted sites like K4 without global changes to DNA and RNA (*Pham et al., 2023*). Therefore, we asked whether changes in histone methylation occurred when intracellular SAM level was inhibited by AurA inhibition. Among the major active histone methylation markers and repressive markers, we found that trimethylation at H3K4 and H3K36, induced by β-glucan stimulation, were decreased by alisertib. Meanwhile, alisertib did not affect trimethylation at H3K9 and monomethylation at H3K4, accompanied with a modest change but no significant difference in trimethylation at H3K27 (*Figure 4A*, *Figure 4—figure supplement 1*). Epigenetic modification including H3K4me3 and H3K36me3 promotes chromatin accessibility thus promoting rapid transcription of genes in response to second stimulation in trained immunity. This prompted us to investigate whether alisertib dampened β-glucan-induced epigenetic modification in genes like *Il6* and *Tnf*. Chromatin immunoprecipitation (ChIP) assays demonstrated that both H3K4me3 and H3K36me3 enrichment on *Il6* and *Tnf* promoters induced by β-glucan were notably decreased by alisertib (*Figure 4B*). Consistently, knockdown of GNMT recovered the expression level of histone methylation and the enrichment level of H3K4me3 as well as H3K36me3 on gene promoter

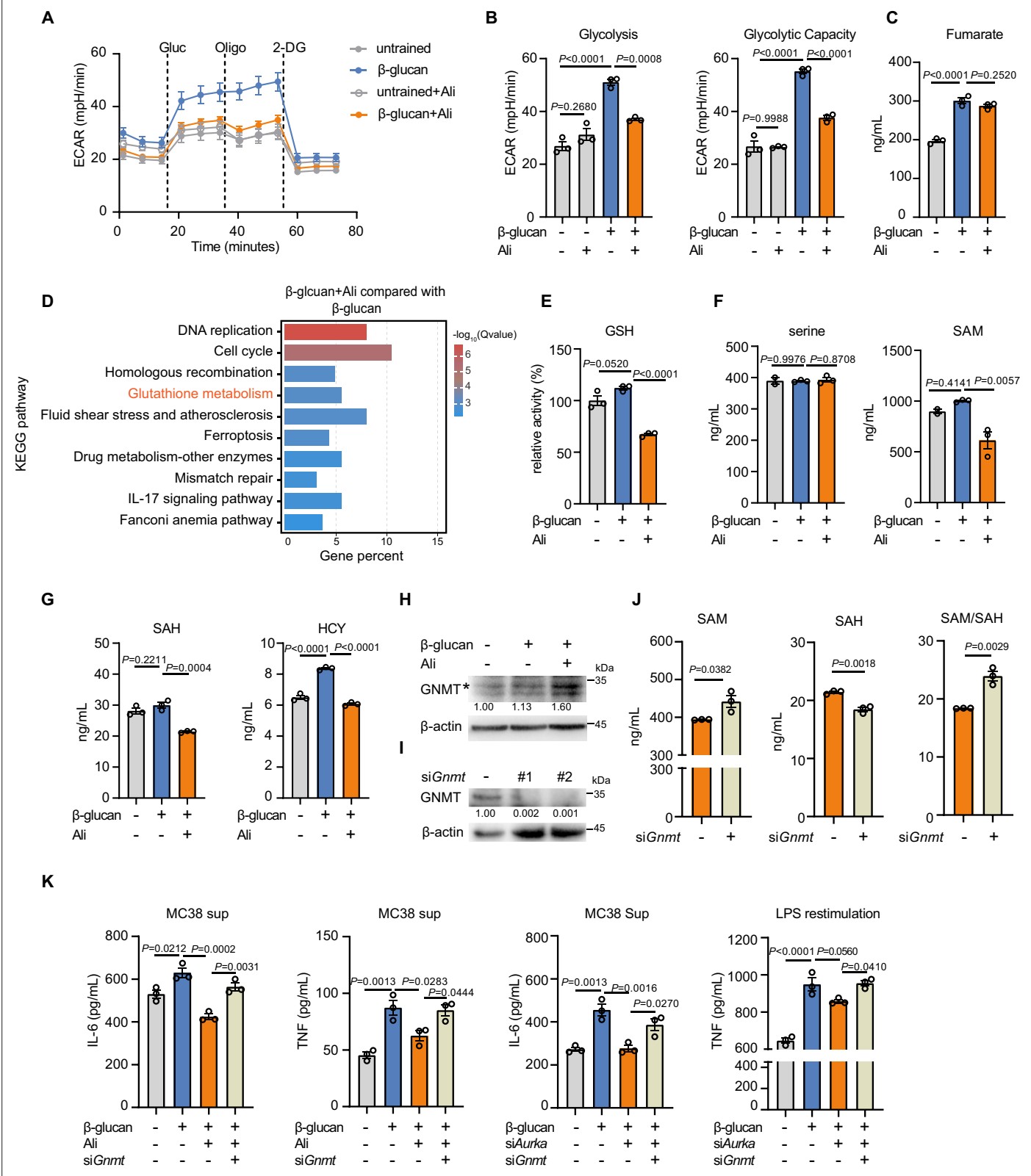

**Figure 3.** Aurora kinase A inhibition decreases glycolysis and S-adenosylmethionine (SAM) level. (**A, B**) Extracellular acidification rate (ECAR) in bone marrow-derived macrophages (BMDMs) with different treatments after a glycolysis stress test upon sequential addition of glucose (Gluc, 10 mM), oligomycin (Oligo, 1 μM), and 2-deoxyglucose (2-DG, 50 mM), as indicated (**A**); basal glycolysis and maximal glycolysis (**B**). (**C, F, G**) LC–MS/MS measurements of fumarate (**C**), serine and SAM (**F**), S-adenosylhomocysteine (SAH) and HCY (**G**) in trained BMDMs treated with vehicle or alisertib for

*Figure 3 continued*

24 hr. (**D**) BMDMs were trained with β-glucan (50 μg/mL) with or without alisertib (1 μM) for 24 hr. The BMDMs were collected for RNA extraction and followed by RNA-seq (n=2 mice per group). The TOP 10 enriched pathways identified by KEGG enrichment analysis of differentially expressed genes (Fold change >1.2, FDR <0.05) by comparing trained BMDMs with or without alisertib. (**E**) Intracellular levels of glutathione in trained BMDMs with or without alisertib for 24 hr. The level was normalized to untrained BMDMs. (**H**) Western blot analysis of GNMT in trained BMDMs treated with vehicle or alisertib for 24 hr. β-actin was used as a loading control; * showed the position of GNMT protein. (**I**) Western blot showing GNMT protein levels in wild-type BMDMs that were transfected with siRNA targeting GNMT. (**J**) LC–MS/MS measurements of SAM and SAH in trained BMDMs treated by alisertib together with or without knockdown of GNMT. The SAM/SAH ratio is calculated by SAH normalization. *P* values were derived from two-tailed t-tests. (**K**) Supernatant levels of IL-6 and TNF in trained BMDMs with AurA inhibition by alisertib or by siRNAs targeting AurA or GNMT. N=3 per group. Data are presented as the mean ± SEM. *P* values were derived from one-way ANOVA with Tukey's multiple-comparison test (**B, K**) compared with every other group or with Dunnett's multiple-comparison test (**C, E–G**) compared with only β-glucan stimulation group. In **H** and **I**, similar results were obtained for three independent experiments. Related to *Figure 3—figure supplement 1*, *Figure 3—source data 1–2*.

The online version of this article includes the following source data and figure supplement(s) for figure 3:

**Source data 1.** Uncropped and labeled blots for *Figure 3H and I*.

**Source data 2.** Raw unedited blots for *Figure 3H and I*.

**Figure supplement 1.** Alisertib inhibits glucose incorporation into glycolysis and tricarboxylic acid (TCA) cycle.

of *Il6* and *Tnf*, which was inhibited by alisertib (*Figure 4B and C*). Therefore, suppression of trained immunity caused by AurA inhibition is a consequence of decreased trimethylation events in histone by SAM deficiency.

## Aurora kinase A regulates GNMT through transcription factor FOXO3

Given the role of GNMT in affecting SAM level, we questioned how AurA inhibition mediated GNMT expression in trained immunity. It was reported that nuclear FOXO3 facilitated GNMT expression to regulate the cell redox response (*Hwang et al., 2021*). Motif analysis of the ATAC-seq indicated that AurA inhibition promoted FOXO signal (*Figure 2C*). Thus, we hypothesized that AurA inhibition upregulated GNMT expression via FOXO3. First, we found that knockdown of FOXO3 downregulated GNMT protein level in BMDMs (*Figure 5A*). Meanwhile, FOXO3 knockdown prevented the enhanced GNMT expression under AurA inhibition and knockdown (*Figure 5B and C*), indicating the involvement of FOXO3 in AurA function. Moreover, knockdown of FOXO3 also restored the trained immunity as indicated by elevation of IL-6 level, which was inhibited by alisertib (*Figure 5D*). To further investigate how AurA regulates FOXO3, we detected the phosphorylation level of FOXO3. The result showed that β-glucan stimulation induced the phosphorylation of FOXO3 at Ser315 and AurA inhibition prevented the phosphorylation induction by β-glucan (*Figure 5E*). Consistently, immunofluorescence detection showed a higher ratio of nuclear enrichment of FOXO3 by AurA inhibition (*Figure 5F*). Nuclear enrichment of FOXO3 is inhibited by AKT-mTOR activation (*Liu et al., 2018*; *van der Vos et al., 2012*). To demonstrate whether the increased FOXO3 nuclear localization by alisertib was associated with AKT-mTOR activation, we checked the activation of AKT-mTOR in trained BMDMs. As expected, both pharmacological and genetic disruption of AurA significantly inhibited β-glucan-induced phosphorylation of AKT-mTOR-S6K-S6 (*Figure 5G*). Additionally, mTOR activation by its agonist MHY1485 promoted trained immunity under aurora kinase A inhibition (*Figure 5H*), suggesting that mTOR-FOXO3 signaling pathway acts downstream of AurA-mediated trained immunity. In conclusion, these results show the role of transcription factor FOXO3 in AurA-mediated GNMT expression in trained immunity, which depends on the activation of mTOR pathway.

## Alisertib abrogates the anti-tumor effect induced by trained immunity

Trained immunity has been implicated in anti-tumor immunity and is considered as a new branch for cancer immunotherapy development (*Bird, 2023*; *Ding et al., 2023*; *Wang et al., 2023b*). We speculated about whether inhibition of AurA would disrupt the anti-tumor effect induced by β-glucan. Therefore, we conducted a tumor model in mice preinjected with β-glucan (*Figure 6A*). Training with β-glucan inhibited MC38 tumor growth, but the administration of alisertib abolished the antitumor effect induced by β-glucan (*Figure 6B*, *Figure 6—figure supplement 1A*). Given that trained immunity mediates long-term effects on myeloid cells via modulation on bone marrow (BM), we performed bone marrow transplantation (BMT) experiment to directly assess whether alisertib acted on BM to inhibit trained immunity in vivo. BM cells were isolated from CD45.1[+] mice, which were either trained

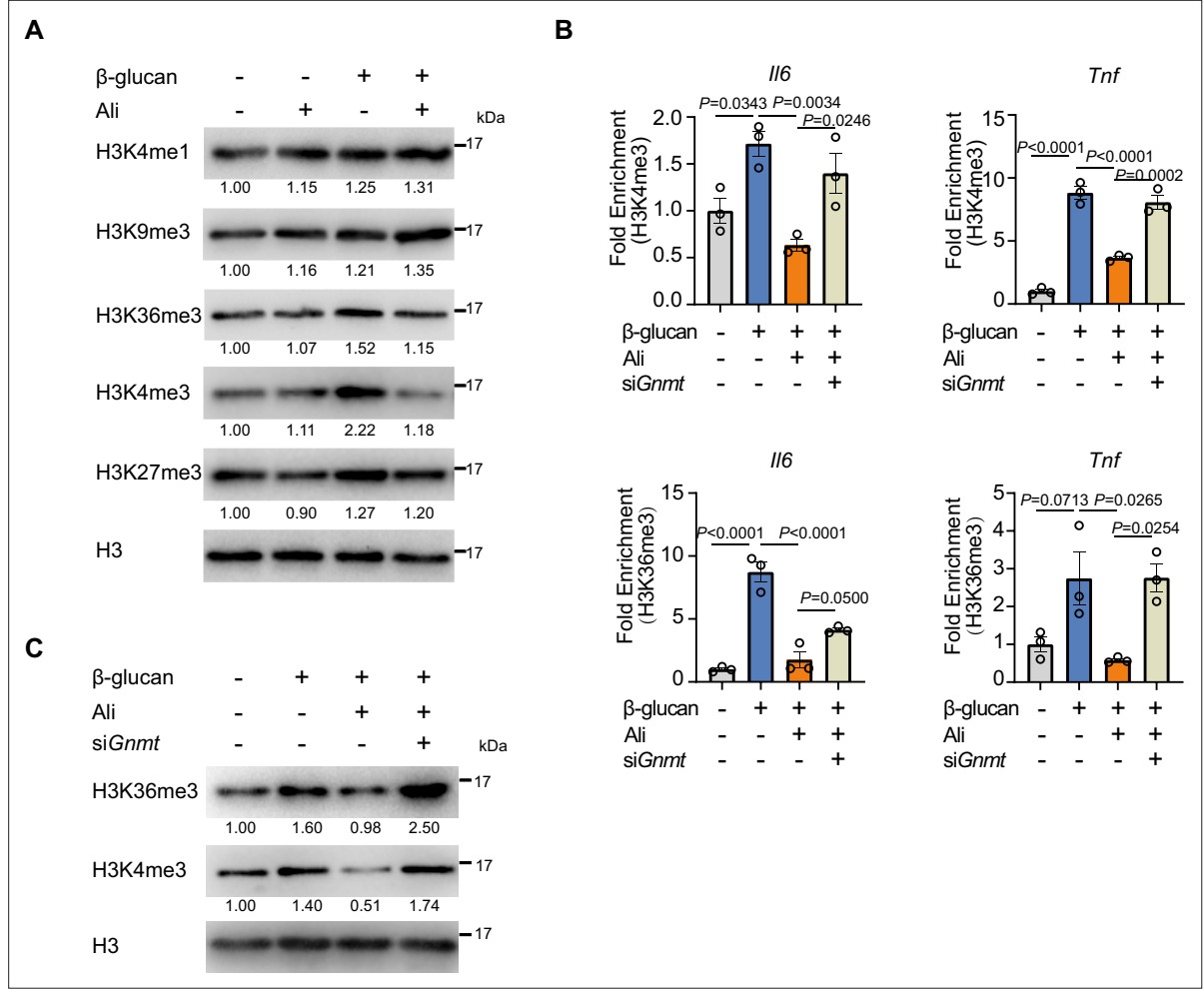

**Figure 4.** Inhibition of Aurora kinase A impairs histone trimethylation at H3K4 and H3K36. (**A**) Western blot analysis of histone methylation levels in trained BMDM treated with vehicles or alisertib. Histone 3 (H3) was used as a loading control. bone marrow-derived macrophages (BMDMs) were trained with β-glucan (50 µg/mL) with or without alisertib (1 µM) for 24 hr, then BMDMs were washed and cultured in fresh medium for 3 days, followed by protein extraction and Western blot analysis. (**B**) ChIP-qPCR analysis of H3K4me3 and H3K36me3 enrichment in *Il6* and *Tnf* promoter regions in trained BMDMs treated with vehicles or alisertib for 24 h and rest for 3 days. N=3 per group. (**C**) Western blot analysis of total H3K4me3 and H3K36me3 levels upon GNMT knockdown in BMDMs. The BMDMs were transfected with siRNA targeting GNMT for 48 hr, followed by β-glucan (50 µg/mL) with or without alisertib (1 µM) treatment for 24 hr. Then the BMDMs were washed and cultured in fresh medium for 3 days and the protein was extracted for Western blot analysis. Data are presented as the mean ± SEM. *P* values were derived from one-way ANOVA with Tukey's multiple-comparison test compared with each other. In **A** and **C**, similar results were obtained for three independent experiments. Related to *Figure 4—figure supplement 1*, *Figure 4—source data 1–2*.

The online version of this article includes the following source data and figure supplement(s) for figure 4:

**Source data 1.** Uncropped and labeled blots for *Figure 4A and C*.

**Source data 2.** Raw unedited blots for *Figure 4A and C*.

**Figure supplement 1.** Inhibition of Aurora kinase A impairs histone trimethylation at H3K4 and H3K36.

by β-glucan with or without alisertib administration, or left untrained, and transferred to lethally irradiated CD45.2$^+$ mice. Four weeks post-BMT, CD45.2$^+$ mice were inoculated with MC38 tumor cells and tumor growth was monitored (*Figure 6C*, *Figure 6—figure supplement 1B*). Compared to mice received BM cells from untrained mice, the tumor burden was significantly decreased in mice that had received BM cells from trained mice, but not in mice that received BM cells from trained mice with alisertib intervention (*Figure 6D*, *Figure 6—figure supplement 1C*). Therefore, the trained immunity-suppressive properties of alisertib can be transferred by BM transplantation to non-trained recipient mice, which confirmed the direct effect of alisertib on BM-derived cells. Next, we investigated the

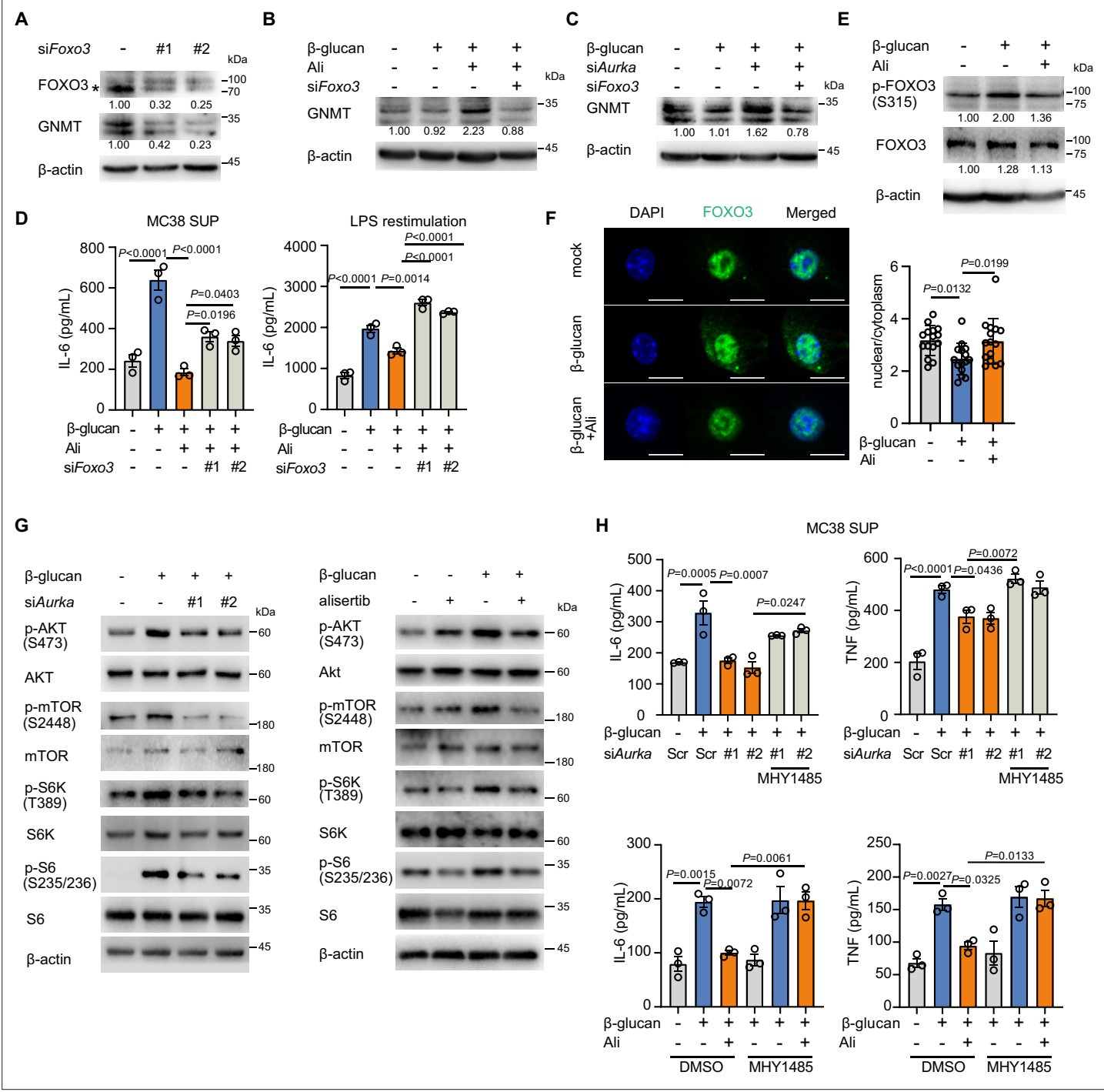

**Figure 5.** Aurora kinase A regulates glycine N-methyltransferase (GNMT) through transcription factor FOXO3. (**A**) Protein level of GNMT in bone marrow-derived macrophages (BMDMs) with FOXO3 knockdown was detected by Western blot; * showed the position of FOXO3 band. (**B, C**) Western blot analysis of GNMT downregulation by si*Foxo3* in trained BMDMs with AurA inhibition. BMDMs were transfected with siRNA targeting FOXO3 for 48 hr, followed by β-glucan (50 μg/mL) and alisertib (1 μM) for 24 hr (**B**); BMDMs were co-transfected with siRNAs targeting FOXO3 and AurA for 48 hr, followed by β-glucan (50 μg/mL) stimulation for another 24 hr (**C**). (**D**) Supernatant levels of IL-6 in BMDMs. The cells were treated with β-glucan (50 μg/mL) and alisertib (1 μM) after transfection of siRNAs targeting FOXO3. (**E**) Western blot analysis of phosphorylation level of FOXO3 at ser 315 in BMDMs treated with β-glucan (50 μg/mL) with or without alisertib (1 μM) for 12 hr. (**F**) Immunofluorescence staining of FOXO3 in BMDMs after 12 hr β-glucan stimulation with or without alisertib. Scale bars: 10 μm (left). The nuclear localization of FOXO3 was compared by calculating the ratio of mean nuclear intensity to cytoplasmic intensity, and the representative data (right) showed the mean intensity of counted macrophages. (**G**) Western blot analysis of AKT-mTOR-S6 pathway in β-glucan-trained BMDM. BMDMs were transfected with siRNA targeting AurA for 48 hr, followed by β-glucan stimulation for

*Figure 5 continued on next page*

*Figure 5 continued*

6 hr (left); BMDM was trained with β-glucan in the absence or presence of alisertib for 6 hr (right). (**H**) Supernatant levels of IL-6 and TNF in BMDMs. The trained cells were treated with siRNA targeting AurA or alisertib in combination with an mTOR agonist, MHY1485 (2 μM), and restimulated with MC38 culture supernatant for 48 hr. Data in **D** and **H** are representative of three independent experiments and presented as the mean ± SEM. *P* values were derived from one-way ANOVA with Tukey's multiple-comparison test compared with each other in **D** and **H**, or with Dunnett's multiple-comparison test in **F** compared with β-glucan only group. In **A-C** and **E-G**, similar results were obtained for three independent experiments. Related to *Figure 5—source data 1–3*.

The online version of this article includes the following source data for figure 5:

**Source data 1.** Uncropped and labeled blots for *Figure 5A–C, E and G*.

**Source data 2.** Raw unedited blots for *Figure 5A–C, E and G*.

**Source data 3.** Original images for *Figure 5F*.

role of macrophages in the anti-tumor effect induced by β-glucan. We analyzed the population of myeloid cells (CD45$^+$CD11b$^+$) as well as the population of myeloid-derived macrophages (CD45$^+$CD11b$^+$F4/80$^+$) in tumor microenvironment. The population of myeloid cells or macrophage induced by β-glucan or by alisertib intervention showed no changes (*Figure 6E*, *Figure 6—figure supplement 1D*). Studies have shown that GNMT is a tumor suppressor gene and its expression is downregulated in tumor tissue (*DebRoy et al., 2013*; *Simile et al., 2022*). Considering that AurA inhibition enhanced GNMT expression in BMDMs in our experiment, we detected GNMT expression in tumor-associated macrophages (TAMs). We found that TAMs from alisertib-treated mice exhibited a higher expression of GNMT, compared with β-glucan-trained mice (*Figure 6F*). Moreover, compared with alisertib-treated mice, the CD45$^+$CD11b$^+$F4/80$^+$ cells in tumor tissue from β-glucan-trained mice showed a higher intracellular phospho-S6, suggesting that AurA inhibition inhibited mTOR activation in TAMs (*Figure 6G*). Pro-inflammatory cytokines induced by β-glucan adjuvant as well as trained immunity can modulate tumor microenvironment and contribute to antitumor immunity (*Sui and Berzofsky, 2024*; *Zhang et al., 2018*). As expected, we observed that β-glucan treatment induced a higher level of IL-1β, IL-6, and IL-12p70 in tumor microenvironment, and the enhanced cytokines production was inhibited by alisertib (*Figure 6H*, *Figure 6—figure supplement 1E*). Moreover, intracellular cytokine staining showed that trained immunity promotes the secretion of IL-6 mainly in infiltrated myeloid cells and macrophages (*Figure 6I*). Taken together, these results demonstrate that AurA inhibition dampens anti-tumor effect of trained immunity via bone marrow cells and reprogrammed the phenotype of tumor-infiltrated myeloid cells including macrophages.

## Discussion

Trained immunity defines the memory ability of innate immune cells in response to second challenge. Although the concept of 'trained immunity' has been proposed since 2011, the detailed mechanisms that regulate trained immunity are still not completely understood (*Netea et al., 2011*). In this study, we identified that AurA was required for trained immunity. AurA deficiency or inhibition decreased IL-6 and TNF production in β-glucan-trained BMDMs upon rechallenge by LPS or supernatant from tumor cells. Moreover, alisertib impaired the anti-tumor effect induced by β-glucan. We also discovered that alisertib induced GNMT expression via mTOR-FOXO3 axis and thus promoted SAM consumption, resulting in decreased SAM/SAH ratio and decreased H3K36me3 and H3K4me3 modification on the promotors of inflammatory genes, highlighting the role of AurA kinase as a hub for methionine metabolism and epigenetics.

S-adenosylmethionine is one of the most ubiquitous metabolites in host. It contains a methyl group, adenosyl group and ACP group, all of which can be transferred by corresponding transferases (*Lee et al., 2023*). It is reported that SAM supplement in mammals is consumed for methylation events on DNA or histone proteins, which affects biological processes like tissue differentiation and gene expression (*Dai et al., 2020*). In macrophages, DNMT transfers methyl group from SAM to DNA and leads to gene repression, such as *Il6* and *Dusp4* (*Ampomah et al., 2022*; *Ji et al., 2019*; *Jung et al., 2020*). However, SAM is also reported to play an active role in primary macrophages to increase IL-1β expression in response to LPS, and its precursor methionine can promote M1 macrophage activation (*Dos Santos et al., 2017*; *Yu et al., 2019*). But whether SAM is involved in trained immunity is not investigated. In our study, AurA inhibition reduced endogenous SAM level in trained BMDMs. It also

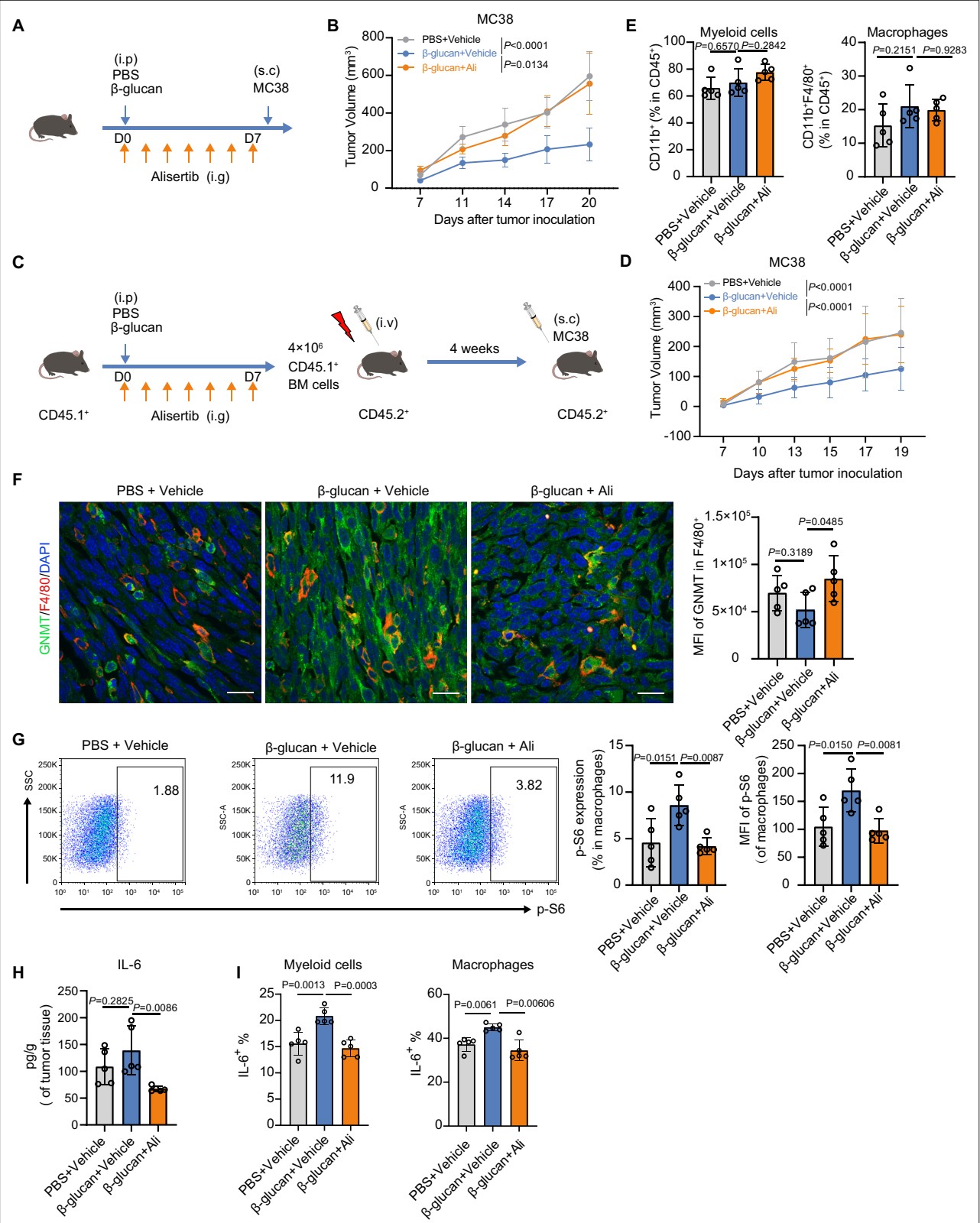

**Figure 6.** Alisertib abrogates the anti-tumor effect induced by trained immunity. (**A**) Experimental scheme of tumor model. 6~8-weeks-old mice was injected with β-glucan together with vehicle or alisertib, followed by subcutaneous inoculation of MC38 cells (1×10⁶ cells/mouse, n=5 per group). (**B**) The MC38 tumor growth curves of mice in A. (**C**) Experimental scheme of bone marrow transplantation model. CD45.1⁺ mice (n=3 per group) as donor were treated as indicated, and CD45.2⁺ mice (n=7 per group) as recipient received 4×10⁶ bone marrow (BM) cells from CD45.1⁺ mice at day 8 post

*Figure 6 continued on next page*

*Figure 6 continued*

irradiation with 9 Gy. After 4 weeks, CD45.2$^+$ mice were inoculated with MC38 cells. (**D**) The MC38 tumor growth curves of mice in C. (**E**) Flow cytometric analysis of myeloid cells (CD45$^+$CD11b$^+$) and macrophages (CD45$^+$CD11b$^+$F4/80$^+$) in MC38 tumors in A. Gating strategy was shown in ***Figure 6—figure supplement 1D***. (**F**) Co-immunofluorescence staining of DAPI, F4/80, and glycine N-methyltransferase (GNMT) in tumor section from A; Scale bars: 20 μm. (**G**) Flow cytometry analysis for intracellular phospho-S6 in macrophages from tumors in A. (**H**) Tumor tissue was lysed and the lysates were collected for detection of IL-6 by ELISA. (**I**) FACS analysis of intracellular IL-6 in tumor-infiltrated myeloid cells and macrophages from tumors in A. Data are represented as the mean ± SD. *P* values were derived from one-way ANOVA (**E, F, G, H, I**) or two-way ANOVA (**B, D**) with Dunnett's multiple-comparison test with data from mice treated with β-glucan only as control. Related to ***Figure 6—figure supplement 1***, ***Figure 6—source data 1***.

The online version of this article includes the following source data and figure supplement(s) for figure 6:

**Source data 1.** Intensity analysis for ***Figure 6F***.

**Figure supplement 1.** Alisertib abrogates the anti-tumor effect induced by trained immunity.

inhibited the trimethylation at H3K4 and histone H3K36 via upregulating GNMT. GNMT is a glycine methyltransferase, which transfers methyl group from SAM to glycine to lower intracellular SAM level (***Luka et al., 2009***). It is well known that inhibition of AurA would cause abnormal chromatin segregation and chromatin instability. Previous research report that chromatin instability make sensitivity to SAM and polyamine-related gene, including GNMT, which might also explain the function of AurA inhibition on SAM (***Islam et al., 2023***). Additionally, SAM is the precursor for glutathione. It has been reported that methionine and SAM reduce oxidative damage by increasing glutathione (***Bandyopadhyay et al., 2022***). However, we found that AurA inhibition didn't increase the level of glutathione, which suggests that alisertib inhibited trained immunity independent of glutathione, but through SAM-mediated epigenetic regulation.

AurA belongs to serine and threonine kinase family. It plays a a critical role in centrosome and chromosome segregation and cell mitosis. Activation of AurA is associated with tumor progression and drug resistance (***Zhao et al., 2023***). During past two decades, there have been developed more than twenty agents as AurA inhibitors (***Attwood et al., 2021***; ***Bavetsias and Linardopoulos, 2015***). These agents have been evaluated across metastatic breast cancer, lung cancer, gastro-oesophageal adenocarcinoma, and peripheral T-cell lymphoma in clinical trials (***Canova et al., 2023***; ***Haddad et al., 2023***; ***Melichar et al., 2015***). Among these inhibitors, alisertib is a highly selective inhibitor for AurA. However, alisertib has limited efficacy against both solid and hematological tumors (***Beltran et al., 2019***; ***Mossé et al., 2019***; ***O'Connor et al., 2019***). In recent years, there are researches trying to explain the resistance to AurA-targeted therapy and reveal feedback loops existed in tumor cells, which contributes to drug resistance. For example, AurA inhibition by alisertib upregulates PD-L1 expression in tumor cells and allows immune escape (***Wang et al., 2023a***). AurA inhibition also enhances fatty acid oxidation to overcome the glycolysis attenuation in tumor cells (***Nguyen et al., 2021***). However, these studies mainly focus on investigating the role of AurA in tumor cells while its role in immune cells is rarely understood. In this study, we identified AurA as a key regulator in promoting trained immunity through epigenetic drug screening. And AurA inhibition significantly decreased the trained immunity induced by β-glucan in vitro and in vivo. AurA is described to be expressed and activated in stem-cell-like cells, bone marrow and epithelium (***Qi et al., 2016***; ***Rio-Vilariño et al., 2024***; ***Zhou et al., 2018***).Clinical data show that alisertib by oral administration rapidly distributes in bone marrow, making bone marrow a site susceptible to side effects (***Oh et al., 2022***). Due to the inhibition effect of alisertib on trained immunity in bone marrow, this may be the potential explanation for the poor therapeutic efficacy or drug resistance of alisertib in patients with cancer. In our primary drug screening, all compounds were examined at a fixed concentration of 5 μM. Despite this limitation, our screening results recapitulated several trained immunity inhibitors previously reported in literature, thereby reinforcing the validity of our screening and highlight some potential molecular targets worthy of further investigation. Overall, our findings provide a potential guidance for AurA inhibitor application in clinical cancer therapy, and the clues for design of new inhibitors such as avoiding targets on bone marrow.

In conclusion, our findings demonstrate that AurA supports trained immunity by maintain SAM levels. As a result of SAM deficiency under AurA inhibition, H3K4m3 and H3K36me3 levels are reduced and trained immunity is inhibited. Our finding reveals a novel AurA-SAM metabolic axis as a new mechanism for trained immunity. Furthermore, our findings also identify a potential guidance

for the clinical application of AurA, and new clues for the design of next-generation AurA inhibitors in the future.

## Materials and methods

### Mice

C57BL/6 J mice (male, 6~8-weeks-old) were purchased from Vital River Laboratory Animal Technology Co., Ltd. (Beijing, China). All mice were kept in specific pathogen-free (SPF) conditions. For all animal studies, mice were randomly assigned to experimental groups, and the sample size was determined according to previous similar study in tumor models (*Arifin and Zahiruddin, 2017*). All animal experiments were performed according to protocols approved by the Institutional Animal Care and Use Committee of Sun Yat-sen University. Mice were injected intraperitoneally with 2 mg β-glucan (Cat# G6513, Sigma-Aldrich, USA) suspended in 200 µL PBS. For drug administration, alisertib (Cat# A4110, ApexBio, Shanghai, dissolved in 10% 2-hydroxypropyl-β-cyclodextrin and 1% sodium bicarbonate, oral gavage) was delivered once daily at 30 mg/kg for 7 days before MC38 inoculation. MC38 ($1 \times 10^6$ cells/mouse) were subcutaneously inoculated into the mice. The tumor volumes were measured every 2 or 3 days, and were calculated as: $1/2 \times$ length $\times$ width$^2$.

### Transplantation and repopulation assay

CD45.1$^+$ B6/SJL male mice (5~6- weeks-old) as donor mice were purchased from Shanghai Model Organisms Center Inc. The mice were injected with β-glucan (2 mg per mouse) or in combination with daily administration of alisertib (30 mg/kg/d) or vehicle for 7 days. On day 8, BM cells were isolated and $4 \times 10^6$ cells were intravenously injected into lethally irradiated CD45.2$^+$ C57BL/6 J male mice (6~8-weeks-old), which received two equal doses of 4.5 Gy at least 4 hr apart with X-ray source (RS2000 Plus-225, RadSource Technologies Inc). CD45.2$^+$ Mice were kept under antibiotic treatment (0.1% levofloxacin) for 1 week before and 4 weeks after irradiation and reconstitution. Four weeks after transplantation, CD45.2$^+$ mice received a subcutaneous injection of MC38 cells ($1 \times 10^6$ cells/mouse). The tumor volumes were measured and recorded every 2 or 3 days.

### Cell culture

THP-1 cells were cultured in 1640 medium with 10% FBS, 1% Pen/Strep, and 1% Glutamax. J774A.1 cells were cultured in DMEM medium with 10% FBS, 1% Pen/Strep. For primary macrophages, bone marrows were isolated from C57BL/6 J mice (6~8 weeks) and cultured in 1640 medium with 10% FBS, 1% Pen/Strep, 10% supernatant from L929 cells. For transfection with siRNAs (synthesized by RuiBiotech, Guangzhou) using RNAiMAX (Invitrogen, Cat# 13778150) according to the manufacturer's instructions, bone marrows cultured for 4 days were digested with trypsin and seeded into cell plates followed by transfection on day 5. Cells were maintained in 95% humidified air and 5% CO2 at 37°C. THP-1 cells and J774A.1 cells were obtained from the Cell Bank of the Chinese Academy of Sciences (Shanghai, China) and have been authenticated by STR profiling. All cells were authenticated tested for mycoplasma contamination.

The siRNA sequences were as follows:

siRNA for mouse AurA#1: 5'-CGAGCAGAGAACAGCUACUUATT-3'
siRNA for mouse AurA#2: 5'-GCACCCUUGGAUCAAAGCUAATT-3'
siRNA for mouse GNMT#1: 5'-GGACAAAGAUGUGCUUUCATT-3'
siRNA for mouse GNMT#2: 5'-CGUCAGUACUGACAGUCAATT-3'
siRNA for mouse FOXO3#1: 5'-CGGCAACCAGACACUCCAATT-3'
siRNA for mouse FOXO3#2: 5'-CUGUAUUCAGCUAGUGCAATT-3'
Scramble siRNA: 5'-UUCUCCGAACGUGUCACGUTT-3'

### Drug screening

Drug screening was performed using a drug library from TargetMol (Shanghai, China). A detailed drugs list was provided in *Supplementary file 1*. For the primary screening, $1 \times 10^5$ BMDMs were seeded into 96-well plates. Following overnight incubation, the BMDMs were trained with β-glucan (100 µg/mL, Sigma-Aldrich, Cat# G6513) in the presence of drugs at a concentration of 5 µM for 24 hr

while the secondary screening for parts of the drugs was performed at two concentrations of 0.2 and 1 μM. Control cells received vehicle control DMSO. After 24 hr of training, cells were washed with 1× PBS and further cultured in fresh complete medium (containing 10% L929 supernatant) for 3 days, followed by LPS stimulation (1 μg/mL). The culture supernatant was subjected to ELISA to measure IL-6. The relative amount of IL-6 was calculated as the fold change of the drugs-treated trained cells compared with the glucan-only-treated cells. The drugs that showed suppressing effects with fold changes of 0.8-fold or lower, were considered to be inhibitors for trained immunity.

## Cell viability assay

The effect of alisertib on cell viability was examined by CCK8 assay (Cat# 40203ES60, Yeasen). In brief, BMDMs were seeded into 96-well plates and trained with β-glucan (50 μg/mL) in the presence of alisertib at different concentrations. After 24 hr treatment, CCK8 reagent was added to the wells for 4 hr, the cell viability was measured at 450 nm.

## Western blot analysis

To evaluate intracellular protein expression, total cellular protein was extracted with lysis buffer containing 50 mM Tris-HCl (pH 7.4), 150 mM NaCl, 1% NP-40, 0.1% SDS, and 0.5% Na-deoxycholate supplemented with Protease Inhibitor Cocktail (Selleck, Cat# B14001) and Phosphatase inhibitors (Roche). Proteins were resolved by SDS-PAGE and subjected to Western blot as described elsewhere. The primary antibodies were incubated at 4°C overnight and HRP-conjugated secondary antibodies was incubated for 1 hr at RT. Enhanced chemiluminescence was used to detect the specific blot bands. The specific bands were quantified using Image Lab (v6.0) and the relative expression was normalized to the internal control.

The following primary antibodies were used: anti-β-actin (1:10000, Cat# A2228, Sigma-Aldrich), anti-Aurora A (1:1000, Cat# ab108353, Abcam), anti-Phospho Aurora A (Thr288) (1:500, Cat# 44–1210 G, Invitrogen), anti-GNMT (1:500, Cat# PA5-100018, Invitrogen), anti-H3K4me1 (1:3000, Cat# ab8895, Abcam), anti-H3K9me3 (1:3000, Cat# ab8898, Abcam), anti-H3K36me3 (1:3000, Cat# ab282572, Abcam), anti-H3K4me3 (1:3000, Cat# ab213224, Abcam), anti-H3K27me3 (1:3000, Cat# 9733, Cell Signaling Technology), anti-H3 (1:5000, Cat# 14269, Cell Signaling Technology), anti-FOXO3 (1:500, Cat# ab23683, Abcam), anti-Phospho FOXO3 (Ser315) (1:500, Cat# 28755–1-AP, Proteintech), anti-AKT (1:1000, Cat# 2920, Cell Signaling Technology), anti-Phospho-AKT (Ser473) (1:1000, Cat# 4060, Cell Signaling Technology), anti-mTOR (1:1000, Cat# 2972, Cell Signaling Technology), anti-Phospho-mTOR (Ser2448) (1:1000, Cat# 5536, Cell Signaling Technology), anti-p70 S6 Kinase (1:1000, Cat# 9202, Cell Signaling Technology), anti-Phospho-p70 S6 Kinase (Thr389) (1:1000, Cat# 9205, Cell Signaling Technology), anti-S6 (1:1000, Cat# 2217, Cell Signaling Technology), and anti-Phospho-S6 (Ser235/236) (1:1000, Cat# 4858, Cell Signaling Technology). Goat anti-rabbit HRP-linked antibody (1:3000, 7074, Cell Signaling Technology), or goat anti-mouse HRP-linked antibody (1:3000, 7076, Cell Signaling Technology) served as the secondary antibody.

## Bulk RNA sequencing

Total RNA was extracted by TRIzol reagent and purified. Library construction and sequencing were performed by Annoroad Gene Technology (Guangzhou). After removal of rRNA and filtering of clean reads, the paired-end reads were mapped to the GRCm38/mm10 genome using HISAT2 2.2.1 (*Florea et al., 2013*). FPKM was obtained for each gene in the RNA-seq data using HTSeq 2.0.2. RNA differential expression analysis was performed by DESeq2 software (*Love et al., 2014*). The genes with FDR (false discovery rate) below 0.05 and absolute fold change (FC) not less than 1.2 were considered as differentially expressed genes (DEGs). All DEGs were mapped to GO terms in the Gene Ontology database for GO enrichment analysis and were mapped to KEGG for pathway enrichment analysis using ClusterProfiler 4.6.6.

## ATAC-seq analysis

$1 \times 10^5$ cells were prepared for ATAC-seq library construction with Novoprotein Chromatin Profile Kit for Illumina according to the manufacturer's instructions. The solubilized DNA fragments were amplified with Novoprotein NovoNGS Index kit for Illumina. The reaction was monitored with qPCR to prevent GC bias and oversaturation. Finally, the library was purified by Novoprotein DNA Clean Beads

and sent to Annoroad Gene Technology (Guangzhou) to sequence with Illumina Novaseq 6000 (150 bp, paired-end). ATAC-seq data were analyzed following guidelines of the Harvard FAS Informatics Group. Optimally, reads were aligned to the reference mouse genome (GRCm38/mm10) using HISAT2 with X, no-spliced-alignment, and no-temp-splice site parameters according to the manual. Quality control was performed with FastQC (Version 0.12.1) in Linux and with ATACseqQC (Version 1.16.0) in R (Version 4.1.0) (*Wilbanks and Facciotti, 2010*). The peaks were called using MACS3 software with default settings tailored for the genome (*Zhang et al., 2008*). For all remaining samples, peaks with low counts (consensus peaks with a median across treatments of counts-per-million less than 1) were filtered out from further analysis and submitted to Diffbind ~3.8.4 to perform further analysis (*Ross-Innes et al., 2012*). The read distribution of PCA plot was generated by dba.plotPCA with the normalized count matrix. And the differential analysis were performed by dba.analyze function with DBA_EDGER. The differentially accessible regions (DARs) were defined as log fold changes >1 or <−1, meanwhile the p<0.05. In order to determine whether differentially accessible chromatin peaks localized near genes with shared functional biological pathways, we performed peak annotation by Genomic Regions Enrichment of Annotations Tool (GREAT) (*Heinz et al., 2010*). We used 500 kb as the maximum absolute distance to the nearest transcriptional start site and q-values less than 0.01 as statistically significant. TF Motif Enrichment Analysis was performed using HOMER's findmotifsGenome. Integrative Genomics Viewer (IGV) track plots of chromatin accessibility were generated with the bigwig files.

## ECAR/OCR measurement and Glutathione detection

Extracellular acidification rate (ECAR) and oxygen consumption rate (OCR) were determined by Seahorse Flux Analyzer XF96 (Agilent) according to the manufacturer's instructions. In brief, BMDMs were plated at a density of $5\times10^4$ cells per well into XFe96 cell culture microplates (Agilent) and cultured overnight followed by β-glucan (50 μg/mL) treatment with or without alisertib (1 μM) for 24 hr. The stimulated cells were then washed with Seahorse XF RPMI Media (pH 7.4) supplemented with 2 mM glutamine and incubated for 1 hr at 37°C without $CO_2$. The cell culture plate was sequentially injected with glucose (10 mM), oligomycin (1 μM), and 2-DG (50 mM) for measurement of ECAR. For OCR measurement, the plate was injected of oligomycin (1 μM), FCCP (1 mM), and rotenone/antimycin A (0.5 μM). Both analyses of ECAR and OCR were performed using an Agilent Seahorse XF96 Analyzer. For intracellular GSH detection, a total of $8\times10^6$ BMDMs were seeded into 10 cm dishes. Cells were trained by β-glucan (50 μg/mL) with or without alisertib (1 μM) for 24 hr. Following that, cells were collected by trypsinization and washed once with 1× PBS. Cell pellet was resuspended in 1 mL of PB buffer before sonication. The lysate was centrifuged at 14,000 rpm at 4°C for 10 min, and supernatants was collected for measurement of glutathione according to the manufacturer's instructions (Cat# DIGT-250, BioAssay Systems). The relative GSH level was calculated by normalization to mock group.

## U13-glucose tracing

Labelled compounds U-$^{13}$C-glucose were added to customized RPMI medium lacking glucose. And BMDMs were trained with β-glucan (50 μg/mL) plus alisertib (1 μM) for 20 hr in this customized medium with labeled U-$^{13}$C-glucose. Following that, cells were washed with 1× PBS twice by aspirating the medium and immediately adding −80°C methanol and pre-cold water containing norvaline. After 20 min of incubation on dry ice, the resulting mixture was scraped, collected into a centrifuge tube, mixed with pre-cold chloroform, and centrifuged at 14,000×g for 5 min at 4°C. The supernatant was evaporated to dryness in a vacuum concentrator. The dry residue was completely dissolved in 20 μL of 2% (w/v) methoxyamine hydrochloride in pyridine and incubated for 60 min at 37°C. N-tert-Butyldimethylsilyl-N-methyltrifluoroacetamide with 1% tert-butyldimethylchlorosilane was used for the preparation of N-tert-butyldimethylsilyl ethanolamines (TBDMs) and sample was incubated with TBDMS for 30 min at 45°C for derivatization. After centrifugation for 3 min at 12,000 rpm, the derived samples were detected and analyzed using a Thermo1310 coupled to an IQ QD MS system with DB-35 (30 m×0.25 mm×0.25 μm). The ion source has an ionization energy of 70 eV and temperature of 300°C. The detection was in a mode of full scan, ranging between 100–650 m/z. The inlet temperature was 270°C, and helium was used as the carrier gas at a flow rate of 1.2 ml/min. Polar metabolite ramp-up procedure: The initial temperature of the column heater was maintained at 100°C for 2 min,

and the temperature was raised to 255°C at a rate of 3.5°C/min and to 320°C at a rate of 15°C/min, with a total run time of approximately 50 min.

## Targeted metabolomics analysis

To measure SAM, SAH, serine, fumarate and HCY, liquid chromatography (LC)–tandem mass spectrometry (MS/MS) analysis was performed. Briefly, BMDMs were trained with β-glucan (50 µg/mL) and inhibited by alisertib for 24 hr. For transfection with siRNAs targeting GNMT, BMDMs were transfected with siRNA for 48 hr and was followed by β-glucan and alisertib treatment for another 24 hr. Metabolites were extracted by using cold 80% methanol (HPLC Grade, Sigma-Aldrich) from 3 million cells with vortex, followed by centrifugation for 15 min at 15,000×g at 4 °C to collect supernatant. The supernatant was then evaporated to dryness in a vacuum concentrator. Analysis of targeted metabolites was conducted on a A6495 triple quadrupole system interfaced with a 1290 UHPLC system (Agilent Technologies). The resulting MS/MS data were processed using Agilent Quantitative Analysis software.

## Chromatin immunoprecipitation (ChIP) assay

The SimpleChIP Enzymatic Chromatin IP Kit (Cat# 9003, Cell Signaling Technology, USA) was used to perform ChIP according to the manufacturer's instructions. Samples were subjected to immunoprecipitation using either Rabbit anti-H3K4me3 antibody, anti-H3K36me3 antibody or a control IgG antibody (Cell Signaling Technology). Fragmented DNAs were purified using spin columns (Axygen) and was used as the templates for qPCR using indicated primer sets spanning the *Tnf* and *Il6* promoters.

The following primers for ChIP-PCR were used:

ChIP *Il6* forward: 5'-TCGATGCTAAACGACGTCACA-3'
ChIP *Il6* reverse: 5'-CGTCTTTCAGTCACTATTAGGAGTC-3'
ChIP *Tnf* forward: 5'-TGGCTAGACATCCACAGGGA-3'
ChIP *Tnf* reverse: 5'-AAGTTTCTCCCCCAACGCAA-3'

## Real-time quantitative PCR

BMDMs were stimulated as indicated and the total RNA was extracted using an EZ-press RNA Purification Kit (EZBioscience) according to the instruments. Then total RNA was reverse-transcribed into cDNA using HiScript III RT SuperMix (Cat# R323, Vazyme), followed by real-time quantitative polymerase chain reaction (qPCR) using 2×PolarSignal SYBR Green mix Taq (Cat# MKG900, MIKX) for analysis of mRNA expression. All data were normalized to *Actb*. Primer sequences were as follows:

Mouse *Actb* forward: 5'-AGAGGGAAATCGTGCGTGAC-3'
Mouse *Actb* reverse: 5'-CAATAGTGATGACCTGGCCGT-3'
Mouse *Il6* forward: 5'-TAGTCCTTCCTACCCCAATTTCC-3'
Mouse *Il6* reverse: 5'-TTGGTCCTTAGCCACTCCTTC-3'
Mouse *Tnf* forward: 5'- CCCTCACACTCAGATCATCTTCT-3'
Mouse *Tnf* reverse: 5'- GCTACGACGTGGGCTACAG-3'

## Immunocytochemistry and immunohistochemistry staining

For intracellular staining of FOXO3 (1:200, Cat# ab23683, Abcam), BMDMs was grown on glass bottom dishes (Nest, Wuxi) overnight. After indicated treatment, BMDMs were washed twice with 1× PBS, fixed in 4% paraformaldehyde in RT for 10 min, permeabilized with 0.25% Triton X-100 for 30 min at RT. Cells were incubated with blocking buffer (1% BSA, 0.1% Tween 20 in TBS) for 30 min at RT. Primary antibodies were incubated at 4 °C overnight, and the appropriate fluorescent secondary antibody (1:500, Cat# A-11008, Thermo Fisher Scientific) for 1 hr at RT. The dishes were then washed three times with 0.1% TBST with DAPI staining during the first wash. For tumor tissue staining, fresh tumor tissue was fixed in 4% paraformaldehyde in RT for 24~48 hr and cut into ~8 µm sections after embedded in OCT. The slice was blocked with normal goat serum with 1% Triton X-100 and incubated with primary antibodies including anti-GNMT (1:200, Cat# PA5-100018, Invitrogen), and anti-F4/80 (1:200, Cat# ab90247, Abcam) antibody overnight at 4 °C, followed by incubation with Alexa Fluor 488 anti-rabbit (1:200, Cat# A-11008, Thermo Fisher Scientific) and Alexa Fluor 555 anti-rat (1:200,

Cat# 4417, Cell Signaling Technology) for 1 h at RT. Nuclei were counterstained with DAPI (1 mg/mL) and mounted with Prolong GLASS antifade mountant (P10144, Thermo Fisher Scientific). Images were obtained using a NIKON confocal microscope. The nuclear localization of FOXO3 in trained BMDMs in vitro was compared by calculating the ratio of mean nuclear intensity to cytoplasmic intensity (*Kelley and Paschal, 2019*). For analysis of the GNMT expression in tumor-associated macrophages, we randomly counted 150~200 DAPI$^+$F4/80$^+$ cells and draw ROI to report intensity. The representative data in our results showed the mean intensity of counted macrophages.

## FACS

Fresh tumor tissue was harvested at the end of time point and cut into small pieces. A single-cell suspension was obtained by crushing the tumor tissue through a 100 μm cell strainer in Phosflow lyse/fix buffer (Cat# 558049, BD Biosciences) immediately. Subsequently, the suspended cells were washed twice with BD perm/wash buffer (Cat# 554723, BD Biosciences). Cells were then stained with the desired antibodies for 30 min at RT in dark. For phospho-S6 staining, cells were incubated with secondary antibody. Cells were analyzed using a BD LSRFortessa flow cytometer.

The following primary antibodies were used: anti-CD45 BV421 (1:200, Cat# 563890, BD Biosciences), anti-CD45.1 FITC (1:200, BioLegend, Cat# 110705), anti-CD45.2 BV510 (1:200, BioLegend, Cat# 109837), anti-CD11b PerCP Cy5.5 (1:200, Cat# 45-0112-82, Thermo Fisher Scientific), anti-F4/80 FITC (1:200, Cat# 11-4801-82, Thermo Fisher Scientific), anti-Phospho-S6 (Ser235/236) (1:200, Cat# 4858, Cell Signaling Technology), and Fixable Viability Dye (1:1000, BioLegend, Cat# 423105). Anti-rabbit IgG (Alexa Fluor 555 Conjugate) (1:200, Cat# 4413, Cell Signaling Technology) served as the secondary antibody.

## Statistical analysis

The GraphPad Prism version 8.0 was used for statistical analysis. Data are presented as mean ± SEM (Standard Error of Mean) of indicated biological replicates for in vitro cellular experiments, while data are presented as mean ± SD (Standard Deviation) of indicated biological replicates for in vivo experiment. For statistical significance analysis, two-tailed t-test was used for comparing two mean values; one-way ANOVA with Dunnett's multiple-comparison test was applied for comparing three or more mean values with β-glucan group as control; one-way ANOVA or two-way ANOVA with Tukey's multiple-comparison test was applied for comparing multiple mean values under different conditions. The data distribution was assumed to be normal, although this assumption was not formally tested.

## Acknowledgements

This work was supported in part by grant (No. 2021YFC2400601) from the National Key R&D Program of China, grant (32370923, 82073140, 82102874, 32400716) from the National Natural Science Foundation of China and grant (QNPG24-03) from Guangzhou Laboratory. We thank the Metabolic Center at Sun Yat-sen University for providing technical support.

## Additional information

### Funding

| Funder | Grant reference number | Author |
| --- | --- | --- |
| National Key Research and Development Program of China | 2021YFC2400601 | Xiaojun Xia<br>Zining Wang |
| National Natural Science Foundation of China | 32370923 | Xiaojun Xia |
| National Natural Science Foundation of China | 82073140 | Zining Wang |
| National Natural Science Foundation of China | 82102874 | Hongxia Zhang |

| Funder | Grant reference number | Author |
| --- | --- | --- |
| National Natural Science Foundation of China | 32400716 | Yongxiang Liu |
| Young Scientists Program of Guangzhou Laboratory | QNPG24-03 | Yongxiang Liu |

The funders had no role in study design, data collection and interpretation, or the decision to submit the work for publication.

## Author contributions

Mengyun Li, Conceptualization, Data curation, Formal analysis, Validation, Investigation, Visualization, Methodology, Writing – original draft, Project administration; Huan Jin, Formal analysis, Investigation, Methodology, Writing – review and editing; Yongxiang Liu, Funding acquisition, Methodology, Writing – review and editing; Zining Wang, Resources, Funding acquisition, Writing – review and editing; Lin Li, Tiantian Wang, Methodology; Xiaojuan Wang, Bitao Huo, Wei Zhao, Jinyun Liu, Peng Huang, Resources; Hongxia Zhang, Resources, Funding acquisition; Tiantian Yu, Shoujie Wang, Formal analysis; Jun Cui, Resources, Funding acquisition, Project administration, Writing – review and editing; Xiaojun Xia, Conceptualization, Resources, Data curation, Supervision, Funding acquisition, Validation, Project administration, Writing – review and editing

## Author ORCIDs

Mengyun Li ⬤ https://orcid.org/0000-0002-1889-6306
Zining Wang ⬤ https://orcid.org/0000-0002-0087-2453
Wei Zhao ⬤ https://orcid.org/0000-0002-0774-2571
Jun Cui ⬤ https://orcid.org/0000-0002-8000-3708
Xiaojun Xia ⬤ https://orcid.org/0000-0003-4444-7472

## Ethics

All Animal experiments were performed according to protocols approved by the Institutional Animal Care and Use Committee of Sun Yat-sen University.

Reviewer #1 (Public review): https://doi.org/10.7554/eLife.104138.3.sa1
Reviewer #2 (Public review): https://doi.org/10.7554/eLife.104138.3.sa2
Author response https://doi.org/10.7554/eLife.104138.3.sa3

# Additional files

## Supplementary files

Supplementary file 1. List of compounds in the epigenetic drug library.

MDAR checklist

Source data 1. Statistical sources data used to generate main figures and figure supplements.

## Data availability

All data generated or analyzed during this study are included in the manuscript and supporting files: source data files have been provided for Figures 1–6 and related figure supplements; statistical source data for figures are provided in *Source data 1*. Raw data of ATAC-Seq and RNA-Seq have been deposited in the NCBI database under accession code GSE280935 and SAMN50519552. The data authenticity of this article has also been validated by uploading the key raw data onto the Research Data Deposit platform (http://www.researchdata.org.cn/) and approved by the Sun Yat-sen University Cancer Center Data Access/Ethics Committee with the approval number RDDB2025703107.

The following datasets were generated:

| Author(s) | Year | Dataset title | Dataset URL | Database and Identifier |
|-----------|------|---------------|-------------|-------------------------|
| Xia X, Li M | 2024 | Aurora kinase A promotes trained immunity via regulation of endogenous S-adenosylmethionine metabolism (house mouse) | https://www.ncbi.nlm.nih.gov/bioproject/?term=GSE280935 | NCBI BioProject, GSE280935 |
| Xia X, Li M | 2025 | Model organism or animal sample from Mus musculus | https://www.ncbi.nlm.nih.gov/biosample/SAMN50519552/ | NCBI BioSample, SAMN50519552 |

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

# Appendix 1

## Appendix 1—key resources table

| Reagent type (species) or resource | Designation | Source or reference | Identifiers | Additional information |
|---|---|---|---|---|
| Chemical compound, drug | β-glucan | Sigma-Aldrich | Cat# G6513 | Inducer of Trained immunity |
| Chemical compound, drug | MHY1485 | MedChemExpress | Cat# HY-B0795 | |
| Chemical compound, drug | (2-Hydroxypropyl)-beta-cyclodextrin | Beyotime | Cat# ST2114 | Solubilizer for oral gavage in animal studies |
| Chemical compound, drug | Lipofectamine RNAiMAX | Invitrogen | Cat# 13778030 | |
| Chemical compound, drug | Tozasertib | TargetMol | Cat# T2509 | |
| Chemical compound, drug | NaHCO3 | Sigma-Aldrich | Cat# S5761 | Solubilizer for oral gavage in animal studies |
| Chemical compound, drug | Alisertib | Apexbio | Cat# A4110 | Aurora kinase A inhibitor; for in vivo administration |
| Chemical compound, drug | Alisertib | TargetMol | Cat# T2241 | Aurora kinase A inhibitor; for in vitro cell culture |
| Cell line (*Homo sapiens*) | THP-1 | Cell Bank of the Chinese Academy of Sciences (Shanghai, China) | CSTR:19375.09.3101 HUMSCSP567 | |
| Cell line (*M. musculus*) | MC38 | Cell Bank of the Chinese Academy of Sciences (Shanghai, China) | CSTR:19375.09.3101 MOUSCSP5431 | |
| Cell line (*M. musculus*) | J774A.1 | Cell Bank of the Chinese Academy of Sciences (Shanghai, China) | CSTR:19375.09.3101 MOUSCSP5224 | |
| Sequence-based reagent (*M. musculus*) | siRNA: nontargeting control | This paper, synthesis by RuiBiotech | | 5'-UUCUCCGAACGUGUCACGUTT-3' |
| Sequence-based reagent (*M. musculus*) | siRNA to AurA#1 | This paper, synthesis by RuiBiotech | | 5'-CGAGCAGAGAACAGCUACUUATT-3' |
| Sequence-based reagent (*M. musculus*) | siRNA to AurA#2 | This paper, synthesis by RuiBiotech | | 5'-GCACCCUUGGAUCAAAGCUAATT-3' |
| Sequence-based reagent (*M. musculus*) | siRNA to GNMT#1 | This paper, synthesis by RuiBiotech | | 5'-GGACAAAGAUGUGCUUUCATT-3' |
| Sequence-based reagent (*M. musculus*) | siRNA to GNMT#2 | This paper, synthesis by RuiBiotech | | 5'-CGUCAGUACUGACAGUCAATT-3' |
| Sequence-based reagent (*M. musculus*) | siRNA to FOXO3#1 | This paper, synthesis by RuiBiotech | | 5'-CGGCAACCAGACACUCCAATT-3' |
| Sequence-based reagent (*M. musculus*) | siRNA to FOXO3#2 | This paper, synthesis by RuiBiotech | | 5'-CUGUAUUCAGCUAGUGCAATT-3' |
| Sequence-based reagent (*M. musculus*) | *Il6*-F | This paper, synthesis by RuiBiotech | PCR primers | 5'-TAGTCCTTCCTACCCCAATTTCC-3' |
| Sequence-based reagent (*M. musculus*) | *Il6*-R | This paper, synthesis by RuiBiotech | PCR primers | 5'-TTGGTCCTTAGCCACTCCTTC-3' |
| Sequence-based reagent (*M. musculus*) | *Tnf*-F | This paper, synthesis by RuiBiotech | PCR primers | 5'- CCCTCACACTCAGATCATCTTCT-3' |
| Sequence-based reagent (*M. musculus*) | *Tnf*-R | This paper, synthesis by RuiBiotech | PCR primers | 5'- GCTACGACGTGGGCTACAG-3' |
| Sequence-based reagent (*M. musculus*) | *Actb*-F | This paper, synthesis by RuiBiotech | PCR primers | 5'-AGAGGGAAATCGTGCGTGAC-3' |

*Appendix 1 Continued on next page*

*Appendix 1 Continued*

| Reagent type (species) or resource | Designation | Source or reference | Identifiers | Additional information |
|---|---|---|---|---|
| Sequence-based reagent (*M. musculus*) | *Actb*-R | This paper, synthesis by RuiBiotech | PCR primers | 5′-CAATAGTGATGACCTGGCCGT-3′ |
| Sequence-based reagent (*M. musculus*) | *Il6*-F | This paper, synthesis by RuiBiotech | Chip-PCR primers | 5′-TCGATGCTAAACGACGTCACA-3′ |
| Sequence-based reagent (*M. musculus*) | *Il6*-R | This paper, synthesis by RuiBiotech | Chip-PCR primers | 5′-CGTCTTTCAGTCACTATTAGGAGTC-3′ |
| Sequence-based reagent (*M. musculus*) | *Tnf*-F | This paper, synthesis by RuiBiotech | Chip-PCR primers | 5′-TGGCTAGACATCCACAGGGA-3′ |
| Sequence-based reagent (*M. musculus*) | *Tnf*-R | This paper, synthesis by RuiBiotech | Chip-PCR primers | 5′-AAGTTTCTCCCCCAACGCAA-3′ |
| Commercial assay or kit | CCK8 | Yeasen | Cat# 40203ES60 | |
| Commercial assay or kit | Protease Inhibitor Cocktail | Selleck | Cat# B14001 | |
| Antibody | Rabbit monoclonal anti-Aurora A | Abcam | Cat# ab108353, RRID:AB_10865712 | IB (1:1000) |
| Antibody | Mouse monoclonal anti-β-actin | Sigma-Aldrich | Cat# A1978, RRID:AB_476692 | IB (1:10000) |
| Antibody | Rabbit polyclonal anti-Phospho Aurora A (Thr288) | Invitrogen | Cat# 44–1210 G | IB (1:500) |
| Antibody | Rabbit polyclonal anti-FOXO3A | Abcam | Cat# ab23683, RRID:AB_732424 | IB (1:500), IF (1:200) |
| Antibody | Rabbit polyclonal anti-Phospho FOXO3 (Ser315) | Proteintech | Cat# 28755–1-AP, RRID:AB_2881210 | IB (1:500) |
| Antibody | Rabbit polyclonal anti-GNMT | Invitrogen | Cat# PA5-100018, RRID:AB_2815548 | IB (1:500), IF (1:200) |
| Antibody | Rabbit monoclonal anti-H3K4me3 | Abcam | Cat# ab213224, RRID:AB_2923013 | IB (1:3000), IP (1:500) |
| Antibody | Rabbit monoclonal anti-H3K4me1 | Abcam | Cat# ab8895, RRID:AB_306847 | IB (1:3000) |
| Antibody | Rabbit polyclonal anti-H3K9me3 | Abcam | Cat# ab8898, RRID:AB_306848 | IB (1:3000) |
| Antibody | Rabbit monoclonal anti-H3K36me3 | Abcam | Cat# ab282572, RRID:AB_3095544 | IB (1:3000), IP (1:500) |
| Antibody | Rabbit monoclonal anti-H3K27me3 | Cell Signaling Technology | Cat# 9733, RRID:AB_2616029 | IB (1:3000) |
| Antibody | Mouse monoclonal anti-H3 | Cell Signaling Technology | Cat# 14269, RRID:AB_2756816 | IB (1:5000) |
| Antibody | Mouse monoclonal anti-AKT | Cell Signaling Technology | Cat# 2920, RRID:AB_1147620 | IB (1:1000) |
| Antibody | Rabbit monoclonal anti- Phospho-Akt (Ser473) | Cell Signaling Technology | Cat# 4060, RRID:AB_2315049 | IB (1:1000) |
| Antibody | Rabbit polyclonal anti-mTOR | Cell Signaling Technology | Cat# 2972, RRID:AB_330978 | IB (1:1000) |
| Antibody | Rabbit monoclonal anti-Phospho-mTOR (Ser2448) | Cell Signaling Technology | Cat# 5536, RRID:AB_1069155 | IB (1:1000) |

*Appendix 1 Continued on next page*

*Appendix 1 Continued*

| Reagent type (species) or resource | Designation | Source or reference | Identifiers | Additional information |
|---|---|---|---|---|
| Antibody | Rabbit polyclonal anti-p70 S6 Kinase | Cell Signaling Technology | Cat# 9202, RRID:AB_331676 | IB (1:1000) |
| Antibody | Rabbit polyclonal anti-Phospho-p70 S6 Kinase (Thr389) | Cell Signaling Technology | Cat# 9205, RRID:AB_330944 | IB (1:1000) |
| Antibody | Rabbit monoclonal anti-S6 | Cell Signaling Technology | Cat# 2217, RRID:AB_331355 | IB (1:1000) |
| Antibody | Rabbit monoclonal anti-Phospho-S6 (Ser235/236) | Cell Signaling Technology | Cat# 4858, RRID:AB_916156 | IB (1:1000), FACS (1:200) |
| Antibody | Rat monoclonal anti-F4/80 | Abcam | Cat# ab90247, RRID:AB_10712189 | IF (1:200) |
| Antibody | Rat monoclonal anti-mouse CD45-BV421 (Brilliant Violet 421) | BD Biosciences | Cat# 563890, RRID:AB_2651151 | FACS (1:100) |
| Antibody | Rat monoclonal anti-mouse CD11b (PerCP-Cyanine5.5) | Thermo Fisher Scientific | Cat# 45-0112-82, RRID:AB_953558 | FACS (1:100) |
| Antibody | Rat monoclonal anti-mouse F4/80 (FITC) | Thermo Fisher Scientific | Cat# 11-4801-82, RRID:AB_2637191 | FACS (1:100) |
| Antibody | Goat anti-Rabbit, Alexa Fluor 488 | Thermo Fisher Scientific | Cat# A-11008, RRID:AB_143165 | IF (1:200) |
| Antibody | Anti-rabbit IgG, Alexa Fluor 555 | Cell Signaling Technology | Cat# 4413, RRID:AB_10694110 | FACS (1:200) |
| Antibody | Anti-rat IgG, Alexa Fluor 555 | Cell Signaling Technology | Cat# 4417, RRID:AB_10696896 | IF (1:200) |
| Commercial assay or kit | IL-6 Mouse Uncoated ELISA Kit | Invitrogen | Cat# 88-7064-88 | |
| Commercial assay or kit | TNF-α Mouse Uncoated ELISA Kit | Invitrogen | Cat# 88-7324-88 | |
| Commercial assay or kit | IL-6 Human Uncoated ELISA Kit | Invitrogen | Cat# 88-7066-88 | |
| Commercial assay or kit | 12p70 Mouse Uncoated ELISA Kit | Invitrogen | Cat# 88-7121-88 | |
| Commercial assay or kit | TGF Mouse Uncoated ELISA Kit | Invitrogen | Cat# 88-8350-88 | |
| Commercial assay or kit | IL-1β Mouse Uncoated ELISA Kit | Invitrogen | Cat# 88-7013-88 | |
| Commercial assay or kit | Glutathione Assay Kit | BioAssay Systems | Cat# DIGT-250 | |
| Commercial assay or kit | Cell Counting Kit (CCK-8) | Yeasen | Cat# 40203ES60 | |
| Commercial assay or kit | Phosflow Lyse/Fix Buffer | BD Biosciences | Cat# 558049 | |
| Commercial assay or kit | BD Perm/Wash | BD Biosciences | Cat# 554723 | |
| Commercial assay or kit | Agilent Seahorse XF Glycolysis Stress Test Kit | Agilent | Cat# 103020–100 | |

*Appendix 1 Continued on next page*

*Appendix 1 Continued*

| Reagent type (species) or resource | Designation | Source or reference | Identifiers | Additional information |
| --- | --- | --- | --- | --- |
| Commercial assay or kit | High-Sensitivity Open Chromatin Profile Kit 2.0 (for Illumina) | Novoprotein | Cat# N248 | |
| Software, algorithm | Fiji | National Institutes of Health | RRID:SCR_002285 | Image analysis |
| Software, algorithm | GraphPad Prism v8.0 | GraphPad Software | RRID:SCR_002798 | Statistical analysis |
| Software, algorithm | FlowJo software v10.8.1 | Tree Star | RRID:SCR_008520 | |
| Other | C57BL/6 Mice | Vital River Laboratory Animal Technology | | |
| Other | CD45.1 B6/SJL Mice | Shanghai Model Organisms | | |

