## [Editor Report · eLife Assessment]

The authors use a range of techniques to examine the role of Aurora Kinase A (AurA) in trained immunity. The study is hypothesis driven, it uses **solid** experimental approaches, and the data are presented in a logical manner. The findings are **valuable** to the trained immunity field because they provide an in-depth look at a common inducer of trained immunity, beta-glucan.

---

## [Referee Report · Reviewer #1 (Public review)]

In this updated and improved manuscript, the authors investigate the role of Aurora Kinase A (AurA) in trained immunity, following a broader drug screening aimed at finding inhibitors of training. They show AurA is important for trained immunity by looking at the different aspects and layers of training using broad omics screening, followed up by a more detailed investigation of specific mechanisms. The authors finalised the investigation with an in vivo MC-38 cancer model where AurA inhibition reduces beta-glucan's antitumour effects.

Strengths:

The experimental methods are generally well-described. I appreciate the authors' broad approach to studying different key aspects of trained immunity (from comprehensive transcriptome/chromatin accessibility measurements to detailed mechanistic experiments). Approaching the hypothesis from many different angles inspires confidence in the results. Furthermore, the large drug-screening panel is a valuable tool as these drugs are readily available for translational drug-repurposing research.

In response to the rebuttal, I would like to compliment and thank the authors for the large amount of work they have done to improve this manuscript. They have removed most of my previous concerns and confusions, and explained some of their approaches in a way that I now agree with them - a great learning opportunity for me as well.

Weaknesses:

(1) The authors have adequately responded to my comments and updated the manuscript accordingly.

(2) The authors have removed most of my concerns. Regarding the use of unpaired tests because that is what is often done in the literature: I still don't agree with this, nor do I think that 'common practice' is a solid argument to justify the approach. However, we can agree to disagree, as I know indeed that many people argue over when paired tests are appropriate in these types of experiments. I appreciate that n=2 for sequencing experiments is justifiable in the way these analyses are used as exploratory screening methods with later experimental validation. I also want to thank the authors for reporting biological replicates where relevant and (I should have mentioned this in my original review also) I appreciate they validate some findings in a separate cell line - many papers neglect this important step.

(3) The authors have adequately responded to my comments and updated the manuscript accordingly.

(4) The authors have adequately responded to my comments and updated the manuscript accordingly.

(5) The authors have adequately responded to my comments and updated the manuscript accordingly.

(6) The authors have adequately responded to my comments and updated the manuscript accordingly. They have actually gone above and beyond.

(7) I would like to thank the authors for highlighting this information and taking away my confusion. The authors have adequately responded to my comments and updated the manuscript accordingly.

(8) The authors have adequately responded to my comments and updated the manuscript accordingly.

(9) I still think adding the 'alisertib alone' control would be of great added value, but I can see how it is unreasonable to ask the authors to redo those experiments.

(10) The authors have adequately responded to my comments and updated the manuscript accordingly.

(11) The authors have adequately responded to my comments and updated the manuscript accordingly.

(12) I thank the authors for their work to repeat this experiment with my suggestions included. I am convinced by this nice data. I would recommend that the authors put the data from New Figure 4 also in the manuscript as it adds value to the manuscript (unless I just missed it, I don't see it in Figure 6 or the supplement). Not every reader may look at the reviewer comments/rebuttal documents.

---

## [Referee Report · Reviewer #2 (Public review)]

Summary:

This manuscript investigates the inhibition of Aurora A and its impact on β-glucan-induced trained immunity via the FOXO3/GNMT pathway. The study demonstrates that inhibition of Aurora A leads to overconsumption of SAM, which subsequently impairs the epigenetic reprogramming of H3K4me3 and H3K36me3, effectively abolishing the training effect.

Strengths:

The authors identify the role of Aurora A through small molecule screening and validation using a variety of molecular and biochemical approaches. Overall, the findings are interesting and shed light on the previously underexplored role of Aurora A in the induction of β-glucan-driven epigenetic change.

Weaknesses:

Given the established role of histone methylations, such as H3K4me3, in trained immunity, it is not surprising that depletion of the methyl donor SAM impairs the training response. Nonetheless, this study provides solid evidence supporting the role of Aurora A in β-glucan-induced trained immunity in murine macrophages. The part of in vivo trained immunity antitumor effect is insufficient to support the final claim as using Alisertib could inhibits Aurora A other cell types other than myeloid cells.

Revision:

The authors have satisfactorily addressed the majority of my concerns. In particular, the new bone marrow transplantation data convincingly demonstrate that Aurora A inhibition with Alisertib abolishes the β-glucan-trained antitumor effect-an essential finding supporting the manuscript's conclusions.

---

## [Author Response]

The following is the authors’ response to the original reviews.

**Reviewer#1 (Public review):**
This work regards the role of Aurora Kinase A (AurA) in trained immunity. The authors claim that AurA is essential to the induction of trained immunity. The paper starts with a series of experiments showing the effects of suppressing AurA on beta-glucan-trained immunity. This is followed by an account of how AurA inhibition changes the epigenetic and metabolic reprogramming that are characteristic of trained immunity. The authors then zoom in on specific metabolic and epigenetic processes (regulation of S-adenosylmethionine metabolism & histone methylation). Finally, an inhibitor of AurA is used to reduce beta-glucan's anti-tumour effects in a subcutaneous MC-38 model.Strengths:With the exception of my confusion around the methods used for relative gene expression measurements, the experimental methods are generally well-described. I appreciate the authors' broad approach to studying different key aspects of trained immunity (from comprehensive transcriptome/chromatin accessibility measurements to detailed mechanistic experiments). Approaching the hypothesis from many different angles inspires confidence in the results (although not completely - see weaknesses section). Furthermore, the large drug-screening panel is a valuable tool as these drugs are readily available for translational drug-repurposing research.

We thank the reviewer for the positive and encouraging comments.

Weaknesses:(1) The manuscript contains factual inaccuracies such as:(a) Intro: the claim that trained cells display a shift from OXPHOS to glycolysis based on the paper by Cheng et al. in 2014; this was later shown to be dependent on the dose of stimulation and actually both glycolysis and OXPHOS are generally upregulated in trained cells (pmid 32320649).

We appreciate the reviewer for pointing out this inaccuracy, and we have revised our statement to ensure accurate and updated description in manuscript. We are aware that trained immunity involves different metabolic pathways, including both glycolysis and oxidative phosphorylation [1, 2]. We also detected Oxygen Consumption Rate (please see response to comment 8 of reviewer#1) but observed no obvious increase of oxygen consumption in trained BMDMs in our experiment setting. As the reviewer pointed out, it might be dependent on the dose of stimulation.

(b) Discussion: Trained immunity was first described as such in 2011, not decades ago.

We are sorry for the inaccurate description, and we have corrected the statement in our revised manuscript as “Although the concept of ‘trained immunity’ has been proposed since 2011, the detailed mechanisms that regulate trained immunity are still not completely understood.”

(2) The authors approach their hypothesis from different angles, which inspires a degree of confidence in the results. However, the statistical methods and reporting are underwhelming.(a) Graphs depict mean +/- SEM, whereas mean +/- SD is almost always more informative. (b) The use of 1-tailed tests is dubious in this scenario. Furthermore, in many experiments/figures the case could be made that the comparisons should be considered paired (the responses of cells from the same animal are inherently not independent due to their shared genetic background and, up until cell isolation, the same host factors like serum composition/microbiome/systemic inflammation etc). (c) It could be explained a little more clearly how multiple testing correction was done and why specific tests were chosen in each instance.

We sincerely thank the reviewer for this thoughtful comment. (a) The data from animal experiments in which trained immunity was induced in vivo are presented as mean ± SD, while the statistical results from cell-based experiments are presented as mean ± SEM in the revised manuscript. (b) We have replaced one-tailed test with two-tailed test (see Figure 3J in revised manuscript, with updated *P* value label). We agree that cells derived from the same animal and subjected to different treatment conditions may be deemed paired data. We reanalyzed our data using paired statistical tests. While this led to a slight reduction in statistical significance for some comparisons, the overall trends remained consistent, and our biological interpretation remains unchanged. For in vitro experiments unpaired statistical tests are commonly used in literature [3, 4]. Thus, we still used unpaired test results here. (c) We have provided a detailed description of how multiple comparisons were performed in revised figure legends.

(d) Most experiments are done with n = 3, some experiments are done with n = 5. This is not a lot. While I don't think power analyses should be required for simple in vitro experiments, I would be wary of drawing conclusions based on n = 3. It is also not indicated if the data points were acquired in independent experiments. ATAC-seq/RNA-seq was, judging by the figures, done on only 2 mice per group. No power calculations were done for the in vivo tumor model.

We are sorry for the confusion in our description in figure legends. For the in vivo experiment, we determined the sample size (n=5, n refers to number of mice used as biological replicates) by referring to the animal numbers used for similar experiments in literatures. And according to a reported resource equation approach for calculating sample size in animal studies [5], n=5-7 is suitable for most of our mouse experiments. The in vitro cell assay was performed at least three independent experiments (BMs isolated from different mice), and each experiment was independently replicated at least three times and points represents biological replicates in our revised manuscript. In Figure 1A, 5 biological replicates of these experiments are presented to carefully determine a working concentration of alisertib that would not significantly affect the viability of trained macrophages, and that was subsequently used in all related cell-based experiments. As for seq data, we acknowledge the reviewer's concern regarding the small sample size (n=2) in our RNA-seq/ATAC-seq experiment. We consider the sequencing experiment mainly as an exploratory/screening approach, and performed rigorous quality control and normalization of the sequencing data to ensure the reliability of our findings. For RNA-seq data analysis, we referred to the DESeq2 manual, which specifies that its statistical framework is based on the Negative Binomial Distribution and is capable of robustly inferring differential gene expression with a minimum of two replicates per group. Therefore, the inclusion of two replicates per group was deemed sufficient for our analysis. Nevertheless, the genomic and transcriptome sequencing data were used primarily for preliminary screening, where the candidates have been extensively validated through additional experiments. For example, we conducted ChIP followed by qPCR for detecting active histone modification enrichment in *Il6* and *Tnf* region to further verify the increased accessibility of trained immunity-induced inflammatory genes.

(e) Furthermore, the data spread in many experiments (particularly BMDM experiments) is extremely small. I wonder if these are true biological replicates, meaning each point represents BMDMs from a different animal? (disclaimer: I work with human materials where the spread is of course always much larger than in animal experiments, so I might be misjudging this.).

Thanks for your comments. In our initially submitted manuscript, some of the statistical results were presented as the representative data (technical replicates) from one of three independent biological replicates (including BMDMs experiments showing the suppression and rescue experiments of trained immunity under different inhibitors or activators, see original Figure 1B-C, Figure 5D, and Figure 5H, also related to Figure 1B-C, Figure 5D, and Figure 5H respectively in our revised manuscript) while other experimental data are biological replicates including CCK8 experiment, metabolic assay and ChIP-qPCR. In response to your valuable suggestion, we have revised the manuscript to present all statistical results as biological replicates from three independent experiments (presented as mean ± SEM), and we have provided all the original data for the statistical analysis results (please see Appendix 2 in resubmit system).

(3) Maybe the authors are reserving this for a separate paper, but it would be fantastic if the authors would report the outcomes of the entire drug screening instead of only a selected few. The field would benefit from this as it would save needless repeat experiments. The list of drugs contains several known inhibitors of training (e.g. mTOR inhibitors) so there must have been more 'hits' than the reported 8 Aurora inhibitors.

Thank you for your suggestion and we have briefly reported the outcomes of the entire drug screening in the revised manuscript. The targets of our epigenetic drug library are primarily categorized into several major classes, including Aurora kinase family, histone methyltransferase and demethylase (HMTs and KDMs), acetyltransferase and deacetylase (HDACs and SIRTs), JAK-STAT kinase family, AKT/mTOR/HIF, PARP family, and BRD family (see New Figure 1, related to Figure 1-figure supplement 1B in revised manuscript). Notably, previous studies have reported that inhibition of mTOR-HIF1α signaling axis suppressed trained immunity[6]. Our screening results also indicated that most inhibitors targeting mTOR-HIF1α signaling exhibit an inhibitory effect on trained immunity. Additionally, cyproheptadine, a specific inhibitor for SETD7, which was required for trained immunity as previously reported [7], was also identified in our screening.

JAK-STAT signaling is closely linked to the interferon signaling pathway, and certain JAK kinase inhibitors also target SYK and TYK kinases. A previous drug library screening study has reported that SYK inhibitors suppressed trained immunity [8]. Consistently, our screening results reveal that most JAK kinase inhibitors exhibit suppressive effects on trained immunity.

BRD (Bromodomain) and Aurora are well-established kinase families in the field of oncology. Compared to BRD, the clinical applications of the Aurora kinase inhibitor are still at early stage. In previous studies using inflammatory arthritis models where trained immunity was established, both adaptive and innate immune cells exhibited upregulated expression of AurA [9, 10]. Our study provides further evidence supporting an essential role of AurA in trained immunity, showing that AurA inhibition leads to the suppression of trained immunity.

(4) Relating to the drug screen and subsequent experiments: it is unclear to me in supplementary figure 1B which concentrations belong to secondary screens #1/#2 - the methods mention 5 µM for the primary screen and "0.2 and 1 µM" for secondary screens, is it in this order or in order of descending concentration?

Thank you for your comments and we are sorry for unclear labelled results in original manuscript (related to Figure 1-supplement 1C). We performed secondary drug screen at two concentrations, and drug concentrations corresponding to secondary screen#1 and #2 are 0.2 and 1 μM respectively. It was just in this order, but not in an order of descending concentration.

(a) It is unclear if the drug screen was performed with technical replicates or not - the supplementary figure 1B suggests no replicates and quite a large spread (in some cases lower concentration works better?)

Thank you for your question. The drug screen was performed without technical replicates for initial screening purpose, and we need to verify any hit in the following experiment individually. Yes, we observed that lower concentration works better in some cases. We speculate that it might be due to the fact that the drug's effect correlates positively with its concentration only within a specific range. But in our primary screening, we simply choose one concentration for all the drugs. This is a limitation for our screening, and we acknowledge this limitation in our discussion part.

(5) The methods for (presumably) qPCR for measuring gene expression in Figure 1C are missing. Which reference gene was used and is this a suitably stable gene?

We are sorry for this omission. The mRNA expression of *Il6* and *Tnf* in trained BMDMs was analyzed by a quantitative real-time PCR via a DDCt method, and the result was normalized to untrained BMDMs with *Actb* (*β-actin*) as a reference gene, a well-documented gene with stable expression in macrophages. We have supplemented the description for measuring gene expression in Material and Methods in our revised manuscript.

(6) From the complete unedited blot image of Figure 1D it appears that the p-Aurora and total Aurora are not from the same gel (discordant number of lanes and positioning). This could be alright if there are no/only slight technical errors, but I find it misleading as it is presented as if the actin (loading control to account for aforementioned technical errors!) counts for the entire figure.

We are very sorry for this omission. In the original data, p-Aurora and total Aurora were from different gels. In this experiment the membrane stripping/reprobing after p-Aurora antibody did not work well, so we couldn’t get all results from one gel, and we had to run another gel using the same samples to blot with anti-aurora antibody and used β-tubulin as loading control for total AurA (please see New Figure 2A, also related to original Figure 1D). We have provided the source data for β-tubulin from the same membrane of total AurA (please see Figure 1-source data). To avoid any potential misleading, we have repeated this experiment and updated this Figure (please see New Figure 2B, also related to Figure 1D in revised manuscript) with phospho-AurA, total AurA and β-actin from the same gel. The bands for phospho AurA (T288) were obtained using a new antibody (Invitrogen, 44-1210G) and we have revised this information in Material and Methods. We have provided data of three biological replicates to confirm the experiment result also see New Figure 2B, related to Figure 1D in revised manuscript, and the raw data have been added in source data for Figure 1.

(7) Figure 2: This figure highlights results that are by far not the strongest ones - I think the 'top hits' deserve some more glory. A small explanation on why the highlighted results were selected would have been fitting.

We appreciate the valuable suggestion. Figure 2 (see also Figure 2 in revised manuscript) presented information on the chromatin landscape affected by AurA inhibition to confirm that AurA inhibition impaired key gene activation involved in pro-inflammatory macrophage activation by β-glucan. In Figure 2B we highlighted a few classical GO terms downregulated including “regulation of growth”, “myeloid leukocyte activation” and “MAPK cascade” (see also Figure 2B in revised manuscript), among which “regulation of growth” is known function of Aurora A, just to show that alisertib indeed inhibited Aurora A function in vivo as expected. “Myeloid leukocyte activation” and “MAPK cascade” were to show the impaired pro-inflammatory gene accessibility. We highlighted KEGG terms downregulated like “JAK-STAT signaling pathway”, “TNF signaling pathway” and “NF-kappa B signaling pathway” in Figure 2F (see also Figure 2F in revised manuscript), as these pathways are highly relevant to trained immunity. Meanwhile, KEGG terms “FOXO signaling pathway” (see also Figure 2G in revised manuscript) was highlighted to confirm the anti-inflammation effect of alisertib in trained BMDMs, which was further illustrated in Figure 5 (see also Figure 5 in revised manuscript, illustrating FOXO3 acts downstream of AurA). Some top hits in Figure 2B like “positive regulation of cell adhesion”, and “pathway of neurodegeneration” and "ubiquitin mediated proteolysis" in Figure 2F and 2G, is not directly related to trained immunity, thus we did not highlight them, but may provide some potential information for future investigation on other functions of Aurora A.

(8) Figure 3 incl supplement: the carbon tracing experiments show more glucose-carbon going into TCA cycle (suggesting upregulated oxidative metabolism), but no mito stress test was performed on the seahorse.

We appreciate this question raised by the reviewer. We previously performed seahorse XF analyze to measure oxygen consumption rate (OCR) in β-glucan-trained BMDMs. The results showed no obvious increase in oxidative phosphorylation (OXPHOS) indicated by OCR under β-glucan stimulation (related to Figure 3-figure supplement 1 A) although the carbon tracing experiments showed more glucose-carbon going into TCA cycle. We speculate that the observed discrepancy between increased glucose incorporation into TCA cycle and unchanged OXPHOS may reflect a characteristic metabolic reprogramming induced by trained immunity. The increased incorporation of glucose-derived carbon into the TCA cycle likely serves a biosynthetic purpose—supplying intermediates for anabolic processes—rather than augmenting mitochondrial respiration[6]. Moreover, the unchanged OXPHOS may be attributed to a reduced reliance on fatty acid oxidation- “catabolism”, with glucose-derived acetyl-CoA becoming the predominant substrate. Thus, while overall OXPHOS remains stable, the glucose contribution to the TCA cycle increases. This is in line with reports showing that trained immunity promotes fatty acid synthesis- “anabolism”[11]. Alternatively, the partial decoupling of the TCA cycle from OXPHOS could result from the diversion of intermediates such as fumarate out of the cycle. Oxygen consumption rate (OCR) after a mito stress test upon sequential addition of oligomycin (Oligo, 1 μM), FCCP (1 mM), and Rotenone/antimycin (R/A, 0.5 μM), in BMDMs with different treatment for 24 h. β-glucan, 50 μg/mL; alisertib, 1 μM.

(9) Inconsistent use of an 'alisertib-alone' control in addition to 'medium', 'b-glucan', 'b-glucan + alisertib'. This control would be of great added value in many cases, in my opinion.

Thank you for your comment. We appreciate that including “alisertib-alone” group throughout all the experiments may further solidify the results. We set the aim of the current study to investigate the role of Aurora kinase A in trained immunity. Therefore, in most settings, we did not include the group of alisertib only without β-glucan stimulation.

(10) Figure 4A: looking at the unedited blot images, the blot for H3K36me3 appears in its original orientation, whereas other images appear horizontally mirrored. Please note, I don't think there is any malicious intent but this is quite sloppy and the authors should explain why/how this happened (are they different gels and the loading sequence was reversed?)

Thank you for pointing out this error. After checking the original data, we found that we indeed misassembled the orientation of several blots in original data submitted. We went through the assembling process and figured out that the orientation of blots in original data was assembled according to the loading sequences, but not saved correctly, so that the orientations in Figure 4A were not consistent with the unedited blot image. We are sorry for this careless mistake, and we have double checked to make sure all the blots are correctly assembled in the revised manuscript. We also provided three replicates of for the Western blot results showing the level of H3K36me3 in trained BMDMs was inhibited by alisertib (as seen in New Figure 7 at recommendation 2 of reviewer#2).

(11) For many figures, for example prominently figure 5, the text describes 'beta-glucan training' whereas the figures actually depict acute stimulation with beta-glucan. While this is partially a semantic issue (technically, the stimulation is 'the training-phase' of the experiment), this could confuse the reader.

Thanks for the reviewer’s suggestion and we have reorganized our language to ensure clarity and avoid any inconsistencies that might lead to misunderstanding.

(12) Figure 6: Cytokines, especially IL-6 and IL-1β, can be excreted by tumour cells and have pro-tumoral functions. This is not likely in the context of the other results in this case, but since there is flow cytometry data from the tumour material it would have been nice to see also intracellular cytokine staining to pinpoint the source of these cytokines.

Thanks for the reviewer’s suggestion. In Figure 6, we performed assay in mouse tumor model and found that trained immunity upregulated cytokines level like IL-6 in tumor tissue, which was downregulated by alisertib administration. In order to rule out the possibility that the detected cytokines such as IL-6 was from tumor cells, we performed intracellular cytokine staining of single cells isolated from tumor tissues (please see New Figure 4). The result showed that only a small fraction of non-immune cells (CD45^-^ population) expressed IL-6 (0.37% ± 0.11%), whereas a significantly higher proportion of IL-6-positive cells was observed among CD45^+^ population (deemed as immune cells, 13.66% ± 1.82%), myeloid cells (CD45^+^CD11b^+^, 15.60% ± 2.19%), and in particular, macrophages (CD45^+^CD11b^+^F4/80^+^37.24% ± 3.04%). These findings strongly suggest that immune cells, especially macrophages, are the predominant source of IL-6 cytokine within the tumor microenvironment. Moreover, we also detected higher IL-6 positive population in myeloid cells and macrophages (please see Figure 6I in revised manuscript).

**Reviewer#2 (Public review):**
Summary:This manuscript investigates the inhibition of Aurora A and its impact on β-glucan-induced trained immunity via the FOXO3/GNMT pathway. The study demonstrates that inhibition of Aurora A leads to overconsumption of SAM, which subsequently impairs the epigenetic reprogramming of H3K4me3 and H3K36me3, effectively abolishing the training effect.Strengths:The authors identify the role of Aurora A through small molecule screening and validation using a variety of molecular and biochemical approaches. Overall, the findings are interesting and shed light on the previously underexplored role of Aurora A in the induction of β-glucan-driven epigenetic change.

We thank the reviewer for the positive and encouraging comments.

Weaknesses:Given the established role of histone methylations, such as H3K4me3, in trained immunity, it is not surprising that depletion of the methyl donor SAM impairs the training response. Nonetheless, this study provides solid evidence supporting the role of Aurora A in β-glucan-induced trained immunity in murine macrophages. The part of in vivo trained immunity antitumor effect is insufficient to support the final claim as using Alisertib could inhibits Aurora A other cell types other than myeloid cells.

We appreciate the question raised by the reviewer. Though SAM generally acts as a methyl donor, whether the epigenetic reprogram in trained immunity is directly linked to SAM metabolism was not formally tested previously. In our study, we provided evidence suggesting the necessity of SAM maintenance in supporting trained immunity. As for in vivo tumor model, we agree that alisertib may inhibits Aurora A in many cell types besides myeloid cells. To further address the reviewer’s concern, we have performed the suggested bone marrow transplantation experiment (trained mice as donor and naïve mice as recipient) to verify the contribution of myeloid cell-mediated trained immunity for antitumor effect (please see New Figure 8, also related to Figure 6C, 6D and Figure 6-figure supplement 1B and 1C in revised manuscript).

**Reviewer #1 (Recommendations for the authors):**
Some examples of spelling errors and other mistakes (by far not a complete list):(a) Introduction, second sentence: reads as if Candida albicans (which should be italicised and capitalised properly) and BCG are microbial polysaccharide components.(b) Methods: ECAR is ExtraCellular Acidification Rate, not 'Extracellular Acid Ratio'(c) Figure 2C: β-glucan is misspelled in the graph title.(d) TNFα has been renamed to 'TNF' for a long time now.(e) Inconsistent use of Tnf and Tfnα (the correct gene symbol is Tnf) (NB: this field does not allow me to italicise gene symbols)(f) Figure supplement 1B: 'secdonary'(g) Caption of figure 4: "Turkey's multiple-comparison test"(h) etcI would ask the authors that they please go over the entire manuscript very carefully to correct such errors.

We apologize for these errors and careless mistakes. We greatly appreciate your suggestions, and have carefully proofread the revised manuscript to make sure no further mistakes.

Please also address the points I raised in the public review about statistical approaches. Even more important than the relatively low 'n' is my question about biological replicates. Please clarify what you mean by 'biological replicate'.If you are able to repeat at least the in vitro experiments (if this is too much work pick the most important ones) a few more times this would really strengthen the results.

Thank you for your comment. Our biological replicates refer to independently repeated experiments using bone marrow cells isolated from different mice, and n represents the number of mice used. We repeated each experiment at least three times using BMDMs isolated from different mice (n = 3, biological replicates). Specifically, we repeated several in vitro experiments showing inhibition of AurA upregulated GNMT in trained BMDMs and showing transcription factor FOXO3 acted as a key protein in AurA-mediated GNMT expression to control trained immunity as well as showing mTOR agonist rescued trained immunity inhibited by alisertib (see New Figure 5, related to Figure 5B-C, Figure 5H in revised manuscript). Additionally, we have provided data with three biological replicates to show the β-glucan induced phosphorylation of AurA (see comment 6 of reviewer#1) and changes of histone modification marker under AurA inhibition and GNMT deficiency (see recommendation 2 of reviewer#2). We also repeated in vivo tumor model to analysis intratumor cytokines (see recommendation 12 of reviewer#1).

Finally: the authors report 'no funders' during submission, but the manuscript contains funding details. Please modify this in the eLife submission system if possible.

Thank you for your kind reminder and we have modified funding information in the submission system.

**Reviewer #2 (Recommendations for the authors):**
(1) I have the following methodological and interpretative comments for consideration:Aurora A has been previously implicated in M1 macrophage differentiation and NF-κB signaling. What is the effect of Aurora A inhibition on basal LPS stimulation? Considering that β-glucan + Ali also skews macrophage priming towards an M2 phenotype, as shown in Fig. 2E, further clarification on this point would strengthen the study.

Thanks for your suggestion. Previous study showed AurA was upregulated in LPS-stimulated macrophages and the inhibition of AurA downregulated M1 markers of LPS-stimulated macrophages through NF-κB pathway but did not affect IL-4-induced M2 macrophage polarization [12]. Consistently, we also found that AurA inhibition downregulated inflammatory response upon basal LPS stimulation as shown by decreased IL-6 level (see New Figure 6). In original Figure 2E (also related to Figure 2E in revised manuscript), we showed an increased accessibility of Mrc1 and Chil3 under “β-glucan +Ali” before re-challenge, both of which are typical M2 macrophage markers. Motif analysis showed that AurA inhibition would upregulate genes controlled by PPARγ (STAT6 was not predicted). Different from STAT6, a classical transcriptional factor in controlling M2 polarization (M2a) dependent on IL-4 or IL-13, PPARγ mediates M2 polarization toward M2c and mainly controls cellular metabolism on anti-inflammation independent on IL-4 or IL-13. Thus, we speculate that inhibition of AurA might promote non-classical M2 polarization, and the details warrant future investigation.

(2) In Figure 4A, it looks like that H3K27me3 is also significantly upregulated by β-glucan and inhibited by Ali. How many biological replicates were performed for these experiments? It would be beneficial to include densitometric analyses to visualize differences across multiple Western blot experiments for better reproducibility and quantitative assessment. In addition, what is the effect of treatment of Ali alone on the epigenetic profiling of macrophages?

We are sorry for this confusion. Each experiment was performed with at least three independent biological replicates. In original Figure 4-figure supplement 1 (also related to Figure 4-figure supplementary 1 in the revised manuscript), we presented the densitometric analysis results from three independent Western blot experiments, which showed that β-glucan did not affect H3K27me3 levels under our experimental conditions. Three biological replicates data for histone modification were shown as follows (New Figure 7, as related to Figure 4-figure supplement 1 in revised manuscript). We appreciate that assay for “Ali alone” in macrophages may add more value to the findings. We set the aim of the current study to investigate the role of Aurora kinase A in trained immunity, and we know that alisertib itself would not induce or suppress trained immunity. Therefore, in most settings, we did not test the effect of Alisertib alone without β-glucan stimulation.

(3) The IL-6 and TNF concentrations exhibit considerable variability (Fig. 3K and Fig. 5H), ranging from below 10 pg/mL to 500-1000 pg/mL. Please specify the number of replicates for these experiments and provide more detail on how variability was managed. Including this information would enhance the robustness of the conclusions.

Thank you for your comment. These experiments were replicated as least three times using BMDMs isolated from different mice. The observed variations in cytokines concentration may be attributed to factors such as differences in cell density, variability among individual mice, and the passage number of the MC38 cells used for supernatant collection. We have prepared new batch of BMDMs and repeated the experiment and provided consistent results in the revised manuscript (please see Figure 5H in revised manuscript). Data for biological replicates have been provided (please see Appendix 2 in resubmit system).

(4) The impact of Aurora A inhibition on β-glucan-induced anti-tumor responses appears complex. Specifically, GNMT expression is significantly upregulated in F4/80- cells, with stronger effects compared to F4/80+ cells as seen in Fig. 6D. To discern whether this is due to the abolishment of trained immunity in myeloid cells or an effect of Ali on tumor cells which inhibit tumor growth, I suggest performing bone marrow transplantation. Transplant naïve or trained donor BM into naïve recipients, followed by MC38 tumor transplantation, to clarify the mechanistic contribution of trained immunity versus off-target effects.

Thanks for your valuable suggestion. Following your suggestion, we have performed bone marrow transplantation to clarify that alisertib acts on the BM cells to inhibit anti-tumor effect induced by trained immunity (see New Figure 8, related to Figure 6C-D in revised manuscript). As the results shown below, transplantation of trained BM cells conferred antitumor activity in recipient mice, while transplantation of trained BM cells with alisertib treatment lost such activity, further demonstrating that alisertib inhibited AurA in trained BM cells to impair their antitumor activity.

References

(1) Ferreira, A.V., et al., Metabolic Regulation in the Induction of Trained Immunity. Semin Immunopathol, 2024. 46(3-4): p. 7.

(2) Keating, S.T., et al., Rewiring of glucose metabolism defines trained immunity induced by oxidized low-density lipoprotein. J Mol Med (Berl), 2020. 98(6): p. 819-831.

(3) Cui, L., et al., N(6)-methyladenosine modification-tuned lipid metabolism controls skin immune homeostasis via regulating neutrophil chemotaxis. Sci Adv, 2024. 10(40): p. eadp5332.

(4) Yu, W., et al., One-Carbon Metabolism Supports S-Adenosylmethionine and Histone Methylation to Drive Inflammatory Macrophages. Mol Cell, 2019. 75(6): p. 1147-1160 e5.

(5) Arifin, W.N. and W.M. Zahiruddin, Sample Size Calculation in Animal Studies Using Resource Equation Approach. Malays J Med Sci, 2017. 24(5): p. 101-105.

(6) Cheng, S.C., et al., mTOR- and HIF-1α-mediated aerobic glycolysis as metabolic basis for trained immunity. Science, 2014. 345(6204): p. 1250684.

(7) Keating, S.T., et al., The Set7 Lysine Methyltransferase Regulates Plasticity in Oxidative Phosphorylation Necessary for Trained Immunity Induced by β-Glucan. Cell Rep, 2020. 31(3): p. 107548.

(8) John, S.P., et al., Small-molecule screening identifies Syk kinase inhibition and rutaecarpine as modulators of macrophage training and SARS-CoV-2 infection. Cell Rep, 2022. 41(1): p. 111441.

(9) Glant, T.T., et al., Differentially expressed epigenome modifiers, including aurora kinases A and B, in immune cells in rheumatoid arthritis in humans and mouse models. Arthritis Rheum, 2013. 65(7): p. 1725-35.

(10) Jeljeli, M.M. and I.E. Adamopoulos, Innate immune memory in inflammatory arthritis. Nat Rev Rheumatol, 2023. 19(10): p. 627-639

(11) Ferreira, A.V., et al., Fatty acid desaturation and lipoxygenase pathways support trained immunity. Nat Commun, 2023. 14(1): p. 7385.

(12) Ding, L., et al., Aurora kinase a regulates m1 macrophage polarization and plays a role in experimental autoimmune encephalomyelitis. Inflammation, 2015. 38(2): p. 800-11.